# Consistent Collaborative Filtering via Tensor Decomposition

**Shiwen Zhao** *swzhao@apple.com*
*Apple*

**Charles G Crissman** *charleyc@gmail.com*
*Work done while at Apple*

**Guillermo R Sapiro** *gsapiro@apple.com*
*Apple*

**Reviewed on OpenReview:** *https://openreview.net/forum?id=HqIuAzBxbh*

## Abstract

Collaborative filtering is the *de facto* standard for analyzing users' activities and building recommendation systems for items. In this work we develop *Sliced Anti-symmetric Decomposition* (SAD), a new model for collaborative filtering based on implicit feedback. In contrast to traditional techniques where a latent representation of users (user vectors) and items (item vectors) are estimated, SAD introduces one additional latent vector to each item, using a novel three-way tensor view of user-item interactions. This new vector extends user-item preferences calculated by standard dot products to general inner products, producing interactions between items when evaluating their relative preferences. SAD reduces to state-of-the-art (SOTA) collaborative filtering models when the vector collapses to 1, while in this paper we allow its value to be estimated from data. Allowing the values of the new item vector to be different from 1 has profound implications. It suggests users may have *nonlinear* mental models when evaluating items, allowing the existence of cycles in pairwise comparisons. We demonstrate the efficiency of SAD in both simulated and real world datasets containing over $1M$ user-item interactions. By comparing with seven SOTA collaborative filtering models with implicit feedbacks, SAD produces the most consistent personalized preferences, in the meanwhile maintaining top-level of accuracy in personalized recommendations. We release the model and inference algorithms in a Python library `https://github.com/apple/ml-sad`.

## 1 Introduction

Understanding preferences based on users' historical activities is key for personalized recommendation. This is particularly challenging when explicit ratings on many items are not available. In this scenario, historical activities are typically viewed as binary, representing whether a user has interacted with an item or not. Users' preferences must be inferred from such *implicit* feedback with additional assumptions based on this binary data.

One common assumption is to view non-interacted items as negatives, meaning users are not interested in them; items that have been interacted are often assumed to be preferred ones (Hu et al., 2008; Pan et al., 2008). In reality however, such an assumption is rarely met. For example, lack of interaction between a user and an item might simply be the result of lack of exposure. It is therefore more natural to assume that non-interacted items are a combination of the ones that users do not like and the ones that users are not aware of (Rendle et al., 2009).

With this assumption, Rendle et al. (2009) proposed to give partial orders between items. Particularly, they assumed that items which users have interacted with are more preferable than the non-interacted ones. With this assumption in mind, Bayesian Personalized Ranking (BPR) was developed to perform personalized

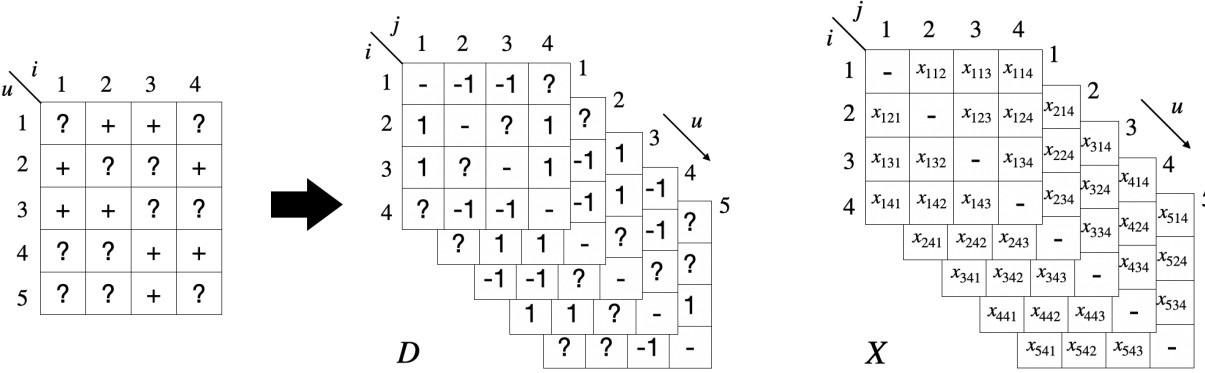

(a) Data transformation to form three-way binary tensor $D$      (b) Tensor parameter $X$

Figure 1: Diagrams provide visualizations of both transformed observation $D$ and parameters $X$ underlying the observation (Rendle et al., 2009).

recommendations. In BPR, the observed data are transformed into a three-way binary tensor $D$ with the first dimension representing users. The other two dimensions represent items, encoding personalized preferences between item pairs (Figure 1a). Mathematically, this means that any first order slice of $D$ at the $u$-th user, $D_{u::}$, is represented as a pairwise comparison matrix (PCM) between items. The $(i, j)$-th entry when observed, $d_{uij} \in \{-1, 1\}$, is binary, suggesting whether $u$-th user prefers ($d_{uij} = 1$) the $i$-th item over the $j$-th one, or the other way around ($d_{uij} = -1$). The tensor $D$ is only partially observed, with missing entries where there is no prior knowledge to infer any preference for a particular user. The recommendation problem becomes finding a parsimonious parameterization of the generative model for observed entries in $D$ and estimating model parameters which best explain the observed data.

The model used in BPR assumes that among the observed entries in $D$, the probability that the $u$-th user prefers the $i$-th item over the $j$-th item can be modeled as a Bernoulli distribution (Hu et al., 2008),

$$p(d_{uij} = 1 | X) = \frac{1}{1 + \exp\left(-x_{uij}\right)}, \tag{1}$$

where $X = \{x_{uij}\} \in \mathbb{R}^{n \times m \times m}$ is the collection of unknown parameters of the Bernoulli distribution (Figure 1b). In fact, $x_{uij}$ is the natural parameter of a Bernoulli distribution. Rendle et al. (2009) decompose the natural parameter by

$$x_{uij} := x_{ui} - x_{uj} \in \mathbb{R}. \tag{2}$$

In the equation, $x_{ui}$ ($x_{uj}$) can be interpreted as the *strength* of preference on the $i$-th ($j$-th) item for the $u$-th user. In other words, users' strengths of preference on different items are represented by scalar values denoted as $x_{ui}$. The relative preferences between items are therefore characterized by the differences between their preference's strengths for a particular user. With this decomposition, $x_{uij}$ becomes $u$-th user's *relative* preference on the $i$-th item over the $j$-th item.

By further letting $x_{ui}$ be represented as a dot product between a $k$ dimensional user vector $\xi_u \in \mathbb{R}^k$ and an item vector $\eta_i \in \mathbb{R}^k$,

$$x_{ui} = \langle \xi_u, \eta_i \rangle, \tag{3}$$

Rendle et al. (2009) reveal the connection between their proposed BPR model and traditional collaborative filtering models such as matrix factorization.

The factorization in equations 2 and 3 provides a parsimonious representation of the original Equation 1. Without the factorization, one could model $x_{uij}$ independently. This model requires $n \times m \times m$ free

parameters, and a model fitting would result in $x_{uij} = \infty$ for $d_{uij} = 1$, and $x_{uij} = -\infty$ for $d_{uij} = -1$. The rests of $x_{uij}$ with missing $d_{uij}$ are left unknown. The parsimonious representation, in contrast, reduces the number of parameters to $(n + m) \times k$. The value of $x_{uij}$ now becomes coupled with $x_{uit}$ for $t \neq j$, and entries in $D$ jointly impact parameter estimate of $x_{uij}$. This highlights a fundamental assumption behind collaborative filtering: information can be learned from items that share similar pattern when they are interacted by users, and one user can learn from another user who interacts with similar set of items.

Equation 3 has a direct link to the Bradley-Terry model often studied when analyzing a PCM for decision making (Hunter, 2004; Weng & Lin, 2011). This model can be at least dated back to 1929 (Zermelo, 1929). One property of the model is its transitivity: The relative preference $x_{uij}$ can be expressed as the sum of relative preferences of $x_{uit}$ and $x_{utj}$, for any user with $t \neq i$ and $t \neq j$. However, in reality this transitivity property is less frequently met. Only around 3% of real world PCM's satisfy complete transitivity (Mazurek & Perzina, 2017). The violation of the property is more conceivable when users are exposed to various types of items. For example, in an online streaming platform, one favorable movie could become less intriguing after a subscriber watches a different style/genre.

In this paper, we extend the original BPR model (which is one of the most fundamental models in recommendation systems) to allow non-transitive user ratings. In particular, we extend Equation 2 to a more general form by proposing a new tensor decomposition. We denote our new model SAD (Sliced Anti-symmetric Decomposition). The new tensor decomposition introduces a second set of non-negative item vectors $\tau_i$ for every item. Different from the first vector $\eta_i$, the new vector contributes negatively when calculating relative preferences, producing counter-effects to the original strength of users' preferences; see Section 4 for more details. Mathematically, the new vector extends the original dot product in Equation 3 to an *inner product*. The original BPR model becomes a special case of SAD when the values in $\tau_i$ are all set to 1, and the inner product reduces to a standard dot product. When $\tau_i$ contains entries that are not 1, the transitivity property no longer necessarily holds. While assigning an $l_1$ regularization to the entries in $\tau_i$ to encourage its values being 1 to reflect prior beliefs, SAD is able to infer its unknown value from real world data. We derive an efficient group coordinate descent algorithm for parameter estimation of SAD. Our algorithm results in a simple stochastic gradient descent (SGD) producing fast and accurate parameter estimations. Through a simulation study we first demonstrate the expressiveness of SAD and efficiency of the SGD algorithm. We then compare SAD to seven alternative SOTA recommendation models in three publicly available datasets with distinct characteristics. Results confirm that our new model permits to exploit information and relations between items not previously considered, and provides more consistent and accurate results as we will demonstrate in this paper.

## 2 Related Works

**Inferring priority via pairwise comparison.** The Bradley-Terry model (Hunter, 2004; Weng & Lin, 2011; Zermelo, 1929) has been heavily used along this line of research. In the Bradley-Terry model, the probability of the $i$-th unit (an individual, a team, or an item) being more preferable than the $j$-th unit (denoted as $i \succ j$) is modeled by

$$p(i \succ j) = \lambda_i/(\lambda_i + \lambda_j), \tag{4}$$

where $\lambda_i$ represents the strength, or degree, of preference of the $i$-th unit. The goal is to estimate $\lambda_i$ for all units based on pairwise comparisons. The link to Equation 1 becomes clear once we omitting user index and set $x_i = \log(\lambda_i)$. In fact, the original BPR model can be viewed as an extension to the Bradley-Terry model to allow personalized parameters, and the strength of preferences are assumed to be dot products of user and item vectors as in Equation 3.

Various algorithms have been developed for parameter estimation of this model. For example, Hunter (2004) developed a class of algorithms named minorization-maximization (MM) for parameter estimation. In MM, a minorizing function $Q$ is maximized to find the next parameter update at every iteration. Refer to the work by Hunter & Lange (2004) for more details related to MM. Weng & Lin (2011) proposed a Bayesian approximation method to estimate team's priorities from outputs of games between teams. Most recently, Wang et al. (2021) developed a bipartite graph iterative method to infer priorities from large and sparse

pairwise comparison matrices. They applied the algorithm to the Movie-Lens dataset to rank movies based on their ratings aggregated from multiple users. Our paper is different from aforementioned models in that we model user-specific item preferences under personalized settings.

**Tensor decompositions for recommendation.** Compared to traditional collaborative filtering methods using matrix factorizations, tensor decompositions have received less attention in this field until recently. The BPR model can be viewed as one of the first attempts to approach the recommendation problems using tensor analysis. As discussed in Section 1, by making the assumption that interacted items are more preferrable compared to non-interacted ones, user-item implicit feedback are represented as a three-way binary tensor (Rendle et al., 2009). In their later work, the authors developed tensor decomposition models for personalized tag recommendation (Rendle & Schmidt-Thieme, 2010). The relationship between their approach and traditional tensor decomposition approaches such as Tucker and PARAFAC (parallel factors) decompositions (Kiers, 2000; Tucker, 1966) was discussed.

Recently, tensor decomposition methods have been used to build recommendation systems using information from multiple sources. Wermser et al. (2011) developed a context aware tensor decomposition approach by using information from multiple sources, including time, location, and sequential information. Hidasi & Tikk (2012) considered implicit feedback and incorporated contextual information using tensor decomposition. They developed an algorithm which scaled linearly with the number of non-zero entries in a tensor. A comprehensive review about applications of tensor methods in recommendation can be found by Frolov & Oseledets (2017) and references therein. Different from leveraging multiple sources of information, the SAD model developed in this paper considers the basic scenario where only implicit feedback are available, the scenario that is considered in BPR model (Rendle et al., 2009). Our novelty lies in the fact that we propose a more general form of tensor decomposition for modeling implicit feedback.

**Deep learning in recommendation models.** Deep learning has attracted significant attention in recent years, and the recommendation domain is no exception. Traditional approaches such as collaborative filtering and factorization machines (FM) have been extended to incorporate neural network components (Chen et al., 2017; He et al., 2017; Xiao et al., 2017). In particular, Chen et al. (2017) replaced the dot product that has been widely used in traditional collaborative filtering with a neural network containing Multilayer Perceptrons (MLP) and embedding layers. Chen et al. (2017) and Xiao et al. (2017) introduced attention mechanisms (Vaswani et al., 2017) to both collaborative filtering and FM (Rendle, 2010). Mostly recently Rendle et al. (2020) revisited the comparison of traditional matrix factorization and neural collaborative filtering and concluded that matrix factorization models can be as powerful as their neural counterparts with proper hyperparameters selected. Despite the controversy, various types of deep learning models including convolutional networks, recurrent networks, variational auto-encoders (VAEs), attention models, and combinations thereof have been successfully applied in recommendation systems. The work by Zhang et al. (2019) provided an excellent review on this topic. This line of research doesn't have direct link to the SAD model considered in the current work. However, we provide a brief review along this line since our model could be further extended to use the latest advances in the area.

## 3 Notation

We use $n$ to denote the total number of users in a dataset and $m$ to denote the total number of items. Users are indexed by $u \in [1, \cdots n]$. Items are indexed by both $i$ and $j \in [1, \cdots m]$. We use $k$ to denote the number of latent factors, and use $h \in [1, \cdots, k]$ to index a factor. Capital letters are used to denote a matrix or a tensor, and lowercase letters to denote a scalar or a vector. For example, the three-way tensor of observations is denoted as $D \in \mathbb{R}^{n \times m \times m}$ and the $(u, i, j)$-th entry is denoted as $d_{uij}$. Similarly, the user latent matrix is denoted as $\Xi \in \mathbb{R}^{k \times n}$, and its $u$-th column is denoted as $\xi_u$ to represent the user vector for $u$-th user. We use $\xi^h$ to denote the $h$-th row (the $h$-th factor) of $\Xi$ viewed as a column vector.

## 4 Tensor Sliced Anti-symmetric Decomposition

We start with the original BPR model. The relative preference $x_{uij}$ defined in Equation 2 forms a three-way tensor $X \in \mathbb{R}^{n \times m \times m}$. The BPR model (Rendle et al., 2009) can be viewed as one parsimonious representation

of tensor $X$. Let $\xi_u \in \mathbb{R}^k$ and $\eta_i \in \mathbb{R}^k$ denote the user and item vectors respectively, and let $\xi_{hu}$ ($\eta_{hi}$) indicate the $h$-th entry in $\xi_u$ ($\eta_i$). Equations 1, 2, and 3 can be re-written as

$$X_{u::} = \sum_{h=1}^{k} \xi_{hu}(\widetilde{H}_h - (\widetilde{H}_h)^\top),$$

where $X_{u::}$ is the first order slice of $X$ at $u$-th user,

$$\widetilde{H}_h = \eta^h \circ \mathbb{1},$$

$\eta^h \in \mathbb{R}^m$ being the $h$-th row of item matrix $H \in \mathbb{R}^{k \times m}$. $\mathbb{1} \in \mathbb{R}^m$ is used to denote a vector of all 1's and $\circ$ being the outer product.

## 4.1 Anti-symmetricity of $X_{u::}$

As discussed in Section 2, $X_{u::}$ represents a parsimonious representation of parameters of a PCM. We formalize the property of $X$ as follows:

*Property* 4.1. For every user $u$, the first order slice of $X$, is an anti-symmetric with $X_{u::} = -X_{u::}^\top$.

This can be shown easily by letting $p_{uij} = p(d_{uij} = 1) = p(i \succ_u j)$ and noting that the relative preference $x_{uij}$ is the natural parameter of the corresponding Bernoulli distribution with $x_{uij} = \log(p_{uij}/(1 - p_{uij}))$. Note that $x_{uij}$ is also known as the log-odds or logit.

The decomposition introduced in BPR can be further written as

$$X_{u::} = \sum_{h=1}^{k} \xi_{hu}(\eta^h \circ \mathbb{1} - \mathbb{1} \circ \eta^h). \tag{5}$$

Note that the anti-symmetricity is well respected in the equation. Intuitively, the decomposition suggests that for the $u$-th user, her item preference matrix $X_{u::}$ can be decomposed as a weighted sum of $k$ anti-symmetric components, each of which is the difference of a rank one square matrix and its transpose.

## 4.2 Generalization of BPR

By replacing $\mathbb{1}$ with arbitrary vector $\tau^h \in \mathbb{R}^m$, the new square matrix $\eta^h \circ \tau^h - \tau^h \circ \eta^h$ is still anti-symmetric, and Property 4.1 still holds for the resulting $X_{u::}$. With this observation, we generalize Equation 5 by proposing a new parsimonious representation of $X_{u::}$,

$$X_{u::} = \sum_{h=1}^{k} \xi_{hu}(\eta^h \circ \tau^h - \tau^h \circ \eta^h). \tag{6}$$

In this work we require entries in $\tau^h$ to be non-negative. The rationale will become clear in Section 4.3. Furthermore, by letting $\Xi := (\xi_1, \xi_2, \cdots, \xi_n) \in \mathbb{R}^{k \times n}$, $H := (\eta^1, \eta^2, \cdots, \eta^k)^\top \in \mathbb{R}^{k \times m}$, and $T := (\tau^1, \tau^2, \cdots, \tau^k)^\top \in \mathbb{R}_+^{k \times m}$ (we use $\mathbb{R}_+$ to denote the set of non-negative real numbers), we introduce the proposed Sliced Anti-symmetric Decomposition (SAD).

**Definition 4.1.** We define the Sliced Anti-symmetric Decomposition (SAD) of $X$ to be the matrices $\Xi, H, T$ satisfying Equation 6 above for every user index $u$. We denote this by

$$X \overset{\text{SAD}}{:=} \{\Xi, H, T\}. \tag{7}$$

## 4.3 Interpretation of SAD

To understand the interpretation of SAD, we start by re-writing equations 2 and 3 in BPR as

$$x_{uij} = \langle \xi_u, \eta_i \rangle - \langle \xi_u, \eta_j \rangle = \sum_{h=1}^{k}(\xi_{hu}\eta_{hi} - \xi_{hu}\eta_{hj}).$$

The term $\xi_{hu}\eta_{hi}$ can be interpreted as the strength of preference of the $u$-th user on the $i$-th item from the $h$-th factor. The overall strength of preference, $x_{ui}$, is the sum of contributions from the $k$ individual factors. Accordingly, the relative preference over the $(i, j)$-th item pair for the $u$-th user can be viewed as the difference between the preference strengths of the $i$-th and the $j$-th items from the $u$-th user.

This interpretation has a direct link to the Bradley-Terry model (4) as previously mentioned, in which the strength of the $i$-th item is described as a positive number $\lambda_i$. Here the strength of preference is viewed as user specific and is represented by a real number $x_{ui} = \log \lambda_{ui}$.

SAD extends the original equations 2 and 3 by introducing a new non-negative vector $\tau_i$ for every item (a column in $T$ in Equation 7). We can re-write Equation 6 as follows for every item pair $i$ and $j$:

$$x_{uij} = \langle \xi_u, \eta_i \rangle_{\text{diag}(\tau_j)} - \langle \xi_u, \eta_j \rangle_{\text{diag}(\tau_i)}$$
$$= \sum_{h=1}^{k} (\xi_{hu}\eta_{hi}\tau_{hj} - \xi_{hu}\eta_{hj}\tau_{hi}). \tag{8}$$

$\langle \cdot, \cdot \rangle_{\text{diag}(\tau_i)}$ in Equation 8 denotes the inner product with a diagonal weight matrix having $\tau_i$ on the diagonal. To be a proper inner product, we require $\tau_i$ to be non-negative, resulting in a positive semi-definite matrix $\text{diag}(\tau_i)$.

The first term on the right hand side of Equation 8, describing the preference strength of the $i$-th item, now becomes dependent on $\tau_{hj}$, the $h$-th entry in $\tau_j$ of the $j$-th rival. When $\tau_{hj}$ is bigger than 1, it increases the effect of $\xi_{hu}\eta_{hi}$. Similarly, the second term on the right hand side suggests that when $\tau_{hi}$ is bigger than 1, it strengthens the effect of the $j$-th item. The opposites happen when either $\tau_{hj}$ or $\tau_{hi}$ is smaller than 1. Therefore, while respecting the anti-symmetricity, the new non-negative item vector $\tau_i$ can be viewed as a counter-effect acting upon the strength of relative preferences, penalizing the strength when greater than 1, while reinforcing when smaller than 1. In real world applications, a user's preference indeed may be influenced by different items. For example, during online shopping, one favorable dress may become less intriguing after a customer sees a different one with different style/color that matches her needs. In an online streaming platform, one favorable movie could become less interesting after a subscriber watches another one with different style/genre. SAD allows us to capture these item-item interactions by introducing a new set of vectors $\tau_i$.

To summarize, we interpret the three factor matrices in SAD as follows:

- $\Xi$ represents the user matrix. Each user is represented by a user vector $\xi_u \in \mathbb{R}^k$.

- $H$ represents the *left* item matrix, which is composed of *left* item vectors denoted as $\eta_i \in \mathbb{R}^k$. It contributes to the strength of preference on the $i$-th item via an inner product with user vector $\xi_u$.

- $T$ represents the *right* item matrix, which contains non-negative *right* item vectors denoted as $\tau_i \in \mathbb{R}_+^k$. This set of vectors defines the weight matrices of inner products between $\eta_i$ and $\xi_u$. It produces counter-effects to the original preference strengths, with values bigger than 1 adding additional strength to rival items in pairwise comparisons, and a value smaller than 1 producing the opposite effect. When $T = 1$, the model reduces to the original BPR model.

In SAD we estimate the value of $T$ from data. As discussed in Section 1, we encourage the values of entries in $T$ to be 1 unless there is strong evidence from the data suggesting otherwise. This is achieved by adding an $l_1$ regularization centered around 1 to the entries in $T$ independently.

The $l_1$ regularization has another side effect. In Equation 8, multiplying by any constant $c$ to $\eta_{hi}$ and $1/c$ to $\tau_{hj}$ results in the same objective function, causing $H$ and $T$ to be unidentifiable. The additional $l_1$ regularization around 1 mitigates the identifiability problem by discouraging any constant multiplication that moves $\tau_{hj}$ away from 1, making the joint objective function identifiable between $H$ and $T$.

### 4.4 The transitivity problem

In social science involving decision makings, PCMs have been investigated extensively (Saaty & Vargas, 2013; Wang et al., 2021). It is usually assumed that a PCM holds the transitivity property, resulting in the following observation introduced in Section 1: The relative preference of the $(i, j)$-th item pair can be derived from the sum of relative preferences of the $(i, t)$-th and $(t, j)$-th item pairs, with $t \neq i$ and $t \neq j$, $x_{uij} = x_{uit} + x_{utj}$. The original BPR meets this property nicely. After introducing $T$ in SAD, this property no longer necessarily holds. One can show that $\tau_i = \tau_j = \tau_t$ for ternary $(i, j, t)$ is a sufficient condition for transitivity in SAD. In our model, we allow the violation of this property, making the proposed model more realistic given the fact that complete transitivity is met only in 3% of real world applications (Mazurek & Perzina, 2017).

### 4.5 Inference algorithms

To estimate model parameters, we maximize the log likelihood function directly. The log likelihood given observed entries in $D$ can be re-written as

$$\log p(D|\Theta) = \sum_{(u,i,j)} \mathbf{1}(d_{uij} = -1)x_{uij} - \log\left(1 + \exp\left(-x_{uij}\right)\right), \tag{9}$$

where $\mathbf{1}(\cdot)$ is the indicator function, and the sum is taken with respect to non-missing entries in $D$ with $i < j$. Here we require $i < j$ to prevent us from double counting.

We take the derivatives with respect to the columns of $\Xi$, $H$, and $T$, resulting in following gradients

$$
\begin{aligned}
\frac{\partial \log p(D|\Theta)}{\partial \xi_u} &= w_{uij}(\eta_i \odot \tau_j - \eta_j \odot \tau_i), \\
\frac{\partial \log p(D|\Theta)}{\partial \eta_i} &= w_{uij}\xi_u \odot \tau_j, \\
\frac{\partial \log p(D|\Theta)}{\partial \eta_j} &= -w_{uij}\xi_u \odot \tau_i, \\
\frac{\partial \log p(D|\Theta)}{\partial \tau_i} &= -w_{uij}\xi_u \odot \eta_j, \\
\frac{\partial \log p(D|\Theta)}{\partial \tau_j} &= w_{uij}\xi_u \odot \eta_i,
\end{aligned}
\tag{10}
$$

from the $(u, i, j)$-th observation. Here $w_{uij} = \mathbf{1}(d_{uij} = -1) + \exp\left(-x_{uij}\right)/(1 + \exp\left(-x_{uij}\right))$ and $\odot$ is the element-wise product (Hadamard product).

Equation 10 allows us to create a stochastic gradient descent (SGD) algorithm (Algorithm 1) to optimize the negative of the log likelihood. During optimization, we add an $l_1$ penalty with weight $w_1$ to the entries in $T$ independently to encourage their values to be 1. In addition, we add an $l_2$ independent penalties with weight $w_2$ to both $\Xi$ and $H$ for further regularization.

We also develop an efficient Gibbs sampling algorithm for full posterior inference under a Probit model setup. By drawing parameter samples from posterior distributions, the Gibbs sampling algorithm has the advantage of producing accurate uncertainty estimation of the unknown parameters under Bayesian inference. We replace the logistic function in Equation 1 with $p(d_{uij} = 1|\Theta) = \Phi(x_{uij})$, where $\Phi(x_{uij})$ is the cumulative distribution function (CDF) of the standard Guassian distribution centered at $x_{uij}$. By assigning spherical Gaussian priors to $\Xi$, $H$, and $T$, full conditional distributions can be derived. More details can be found in Appendix.

## 5 Simulation Study

We first evaluate the performance of SAD and the SGD algorithm on simulation data, with the goal of examining the performance of our algorithm with true parameters known ahead. We choose $n = 20$ users, $m = 50$ items, and $k = 5$, resulting in $\Xi \in \mathbb{R}^{5\times20}$, $H \in \mathbb{R}^{5\times50}$, and $T \in \mathbb{R}_+^{5\times50}$. We consider two scenarios in

---

**Algorithm 1** SGD for parameter estimation of SAD

---

**Require:** $n$, $m$, $k$, $D \in \mathbb{R}^{n \times m \times m}$, $\rho$, $w_1$, $w_2$       $\triangleright$ $\rho$: learning rate, $w_1$, $w_2$ weights for $l_1$ and $l_2$
    Initialization $\Xi \in \mathbb{R}^{k \times n}, H \in \mathbb{R}^{k \times m}, T \in \mathbb{R}_+^{k \times m}$
    $X = \{\Xi, H, T\}$                                                 $\triangleright$ Equation 7
    **while** Convergence not met **do**
       **for** $u = 1 \cdots n$ **do**
          **for** Every item $i$ in interacted set **do**
             Random select item $j$ from non-interacted item set
             Calculate $\mathrm{d}\xi_u$, $\mathrm{d}\eta_i$, $\mathrm{d}\eta_j$, $\mathrm{d}\tau_i$, $\mathrm{d}\tau_j$                     $\triangleright$ Equation 10
             $\xi_u \leftarrow \xi_u + \rho \cdot (\mathrm{d}\xi_u - 2w_2\xi_u)$
             $\eta_i \leftarrow \eta_i + \rho \cdot (\mathrm{d}\eta_i - 2w_2\eta_i)$
             $\eta_j \leftarrow \eta_j + \rho \cdot (\mathrm{d}\eta_j - 2w_2\eta_j)$
             $\tau_i \leftarrow \tau_i + \rho \cdot (\mathrm{d}\tau_i - w_1\mathbf{1}[\tau_i > 1] + w_1\mathbf{1}[\tau_i < 1])$     $\triangleright$ $\mathbf{1}[\cdot]$: Entry-wise indicator to $\tau_i \in \mathbb{R}^k$
             $\tau_j \leftarrow \tau_j + \rho \cdot (\mathrm{d}\tau_j - w_1\mathbf{1}[\tau_j > 1] + w_1\mathbf{1}[\tau_j < 1])$
         **end for**
       **end for**
    **end while**

---

the simulation. In the first simulation (Sim1) we set $T$ to 1, effectively reducing SAD to the generative model of BPR (Rendle et al., 2009). In the second scenario (Sim2), we set a small proportion of $T$ to either 0.01 or 5, the other entries are set to 1. For user matrix $\Xi$ and left item matrix $H$, their values are uniformly drawn from the interval $[-2, 2]$. We calculate the preference tensor $X$ with Equation 8 and draw an observation tensor $D$ from the corresponding Bernoulli distributions.

We first examine the performance of SAD with complete observations in Sim2 to validate if our method is able to generate accurate parameter estimation. We run the SGD algorithm with a learning rate 0.05. The weight of the $l_1$ regularization assigned to $T$ is set to 0.01, and the weight of the $l_2$ regularization is set to 0.005. Initial values of parameters are randomly drawn from a standard Gaussian distribution. The number of latent factors $k$ is set to the true value. In reality when $k$ is unknown, cross validation can be used to select the best value of $k$. After 20 epochs, $\hat{T}$ is able to recover the sparse structure of the true parameters of $T$ up to permutation of factors (Figure 2). The user matrix and left item matrix converge to the true parameter values as well (Figure 3).

Next we examine the performance of SAD in both Sim1 and Sim2, under the scenarios with missing data. To be more specific, we randomly mark $x\%$ of $D$ as missing to mimic missing at random. Note that in the real world, observations could have more complex missingness structures. As a comparison, we run BPR under the same contexts.

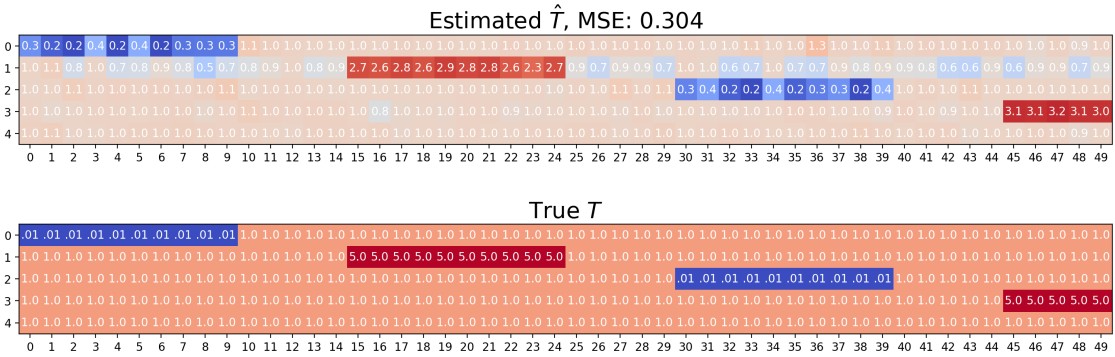

Figure 2: Comparison of $\hat{T}$ with ground truth. Factors are re-ordered in $\hat{T}$ to match true $T$.

The convergences of SAD in Sim1 are shown in Figure 4a, together with the estimated sparsity of $T$. Here the sparsity of $T$ is defined as the percentage of the entries in $T$ with $|\tau_{hj} - 1| < 0.05$. When the percentage

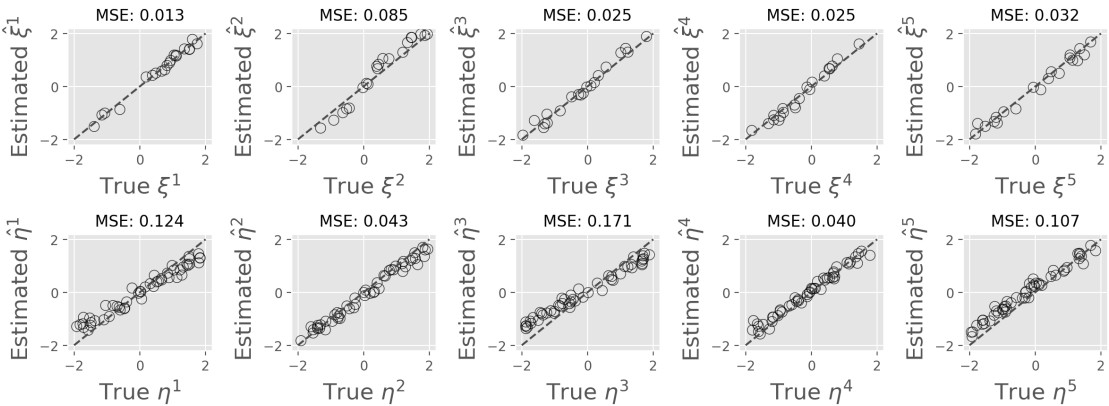

Figure 3: Comparison of $\hat{\Xi}$, $\hat{H}$ with their ground truth. Factors are subject to re-order and sign flips.

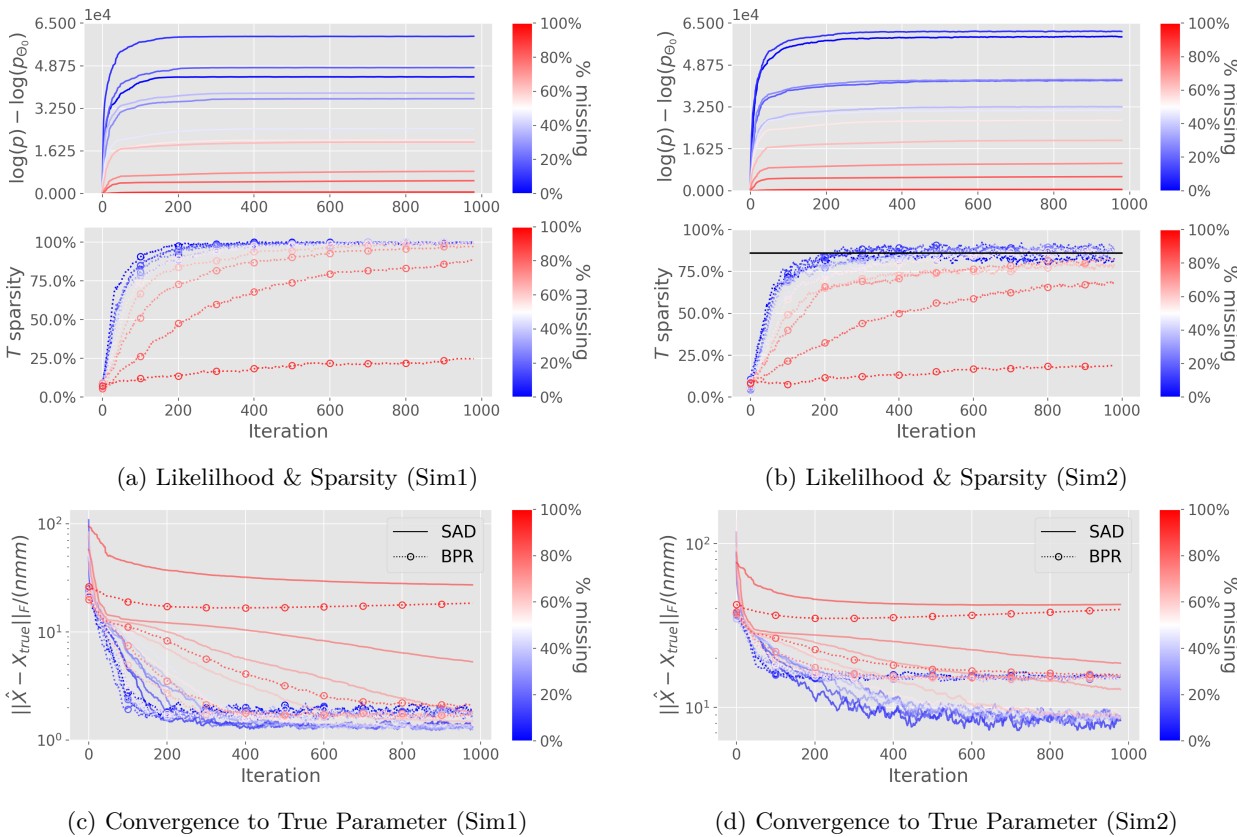

(a) Likelilhood & Sparsity (Sim1)

(b) Likelilhood & Sparsity (Sim2)

(c) Convergence to True Parameter (Sim1)

(d) Convergence to True Parameter (Sim2)

Figure 4: Convergence in Simulation Study. Top rows in 4a and 4b show changes of log likelihood (Equation 9) during SGD optimization. $\log(p_{\Theta_0})$ is the log likelihood at 0-th iteration. Bottom row of 4b shows the true sparsity (86%) as black line. 4c and 4d show the Frobenius distance between estimated parameter $\hat{X}$ and true parameter $X_{true}$ during SGD optimization. Note that SAD (solid line) achieves a much lower distance compared with BPR (dashed line) in 4d under wide range of percentages of missingness.

of missingness is at small or medium levels, SAD is able to converge to a sparsity close to 1, suggesting the effectiveness of the $l_1$ regularization. It becomes more challenging when the percentage of missingness surpasses 70%. Figure 4c shows the trajectories of the mean squared error (MSE) between $\hat{X}$ and the true $X$ under different missingness percentages for both SAD and BPR. Both SAD and BPR are able to converge to

a low MSE with small/medium percentages of missingness. Similarly, with a high percentage of missingness, the performance of both models begins to deteriorate. We conclude that SAD has a performance *on par* with BPR when data are simulated from the generative model of BPR.

Results for Sim2 are shown in Figure 4b. Note that the true sparsity of $T$ is 86%. SAD is able to generate an accurate estimation of the sparsity under small/medium percentages of missingness. When evaluating both models using MSE, SAD is able to achieve a much lower value due to its correct specification of the generative model (Figure 4d).

# 6  Applications for Real Data

We select three real world datasets to evaluate SAD and compare against SOTA recommendation models. The datasets selected contain explicit integer valued ratings. Nonetheless, we mask their values and view them as binary. The explicit ratings are used as a means to evaluate models' consistency in pairwise comparison after model fitting (details below). The first dataset used is from the Netflix Prize (Bennett et al., 2007). The original dataset contains movie ratings of $8,921$ movies from $478,533$ unique users, with a total number of ratings reaching to over $50M$. We randomly select $10,000$ users as our first dataset. The resulting dataset contains $8,693$ movies with over $1M$ ratings from the $10,000$ users. For the second dataset, we choose the Movie-Lens 1M dataset (Harper & Konstan, 2015). It contains over $1M$ ratings from $6,040$ users on $3,706$ movies. As a third dataset, we consider the reviews of recipes from Food-Com (Majumder et al., 2019). The complete dataset has $1.1M$ reviews from $227K$ users on $231K$ recipes. We select the top $20K$ users with the most activities, and filter out recipes receiving less than 50 reviews. The resulting dataset has $145K$ reviews from $17K$ users (users with zero activity are further removed after filtering recipes) on $1.4K$ recipes.

The three datasets have distinct characteristics. Among the three datasets, the Food-Com review dataset has the least user-item interactions, even when the most active users/popular recipes are selected. The maximum number of items viewed by a single user is 878. It also has the largest number of users. The Netflix dataset is the most skewed, with the number of items interacted by a user ranging from as low as 1 to as high as over $8K$. The Movie-Lens dataset contains the largest number of user-item interactions, and is most uniformly distributed. Some details of the three datasets can be found in Table A in Appendix.

We choose seven SOTA recommendation models to compare with SAD. Their details are listed in Table 3. For each of the model considered, we perform a grid search to determine hyperparameters. Models are chosen based on their goodness-of-fit using log likelihood. We evaluate the models using a comprehensive leave one interaction out (LOO) evaluation (Bayer et al., 2017; He et al., 2017; 2016), in which we randomly hold out one user-item interaction from training set for every user. Users who have only one interaction are skipped. We create 20 such LOO sets for each dataset considered. The dimension of latent space during evaluation is set to 500 for all models and datasets. The choice of the latent dimension can be further optimized using methods such as across validation. In this work we choose the same number across comparing SOTA models such that they can be evaluated on a common ground.

$$\frac{1}{n \times (|I_u| - 1)} \sum_u \sum_{j \in I_u, j \neq o} \left( \hat{x}_{hoj} \mathbf{1}(o \succ j) + \hat{x}_{hjo} \mathbf{1}(j \succ o) \right) \tag{11}$$

$$\frac{1}{n \times (|I_u| - 1)} \sum_u \sum_{j \in I_u, j \neq o} \left( \mathbf{1}(\hat{x}_{hoj} > 0)\mathbf{1}(o \succ j) + \mathbf{1}(\hat{x}_{hjo} > 0)\mathbf{1}(j \succ o) \right) \tag{12}$$

We consider two aspects of model's performance during evaluation: consistency and recommendation. The consistency is defined as whether model's prediction matches user's actual pairwise preference. During evaluation, the hold out item $o$ and other items $j$ in interacted set $I_u$ of user $u$ are arranged such that $o \succ_u j$ based on users' actual ratings. The mean of their predicted preference (Equation 11), the percentage of consistent predictions (Equation 12), and the median of per user percentage of consistency are reported in Table 1. SAD has the most consistent results among all eight recommendation models except in one scenario, in which our model is second best.

Table 1: Model evaluations across 20 LOO datasets. When evaluating consistency, item pairs between hold out item $o$ and other interacted items $j$ for a user are ordered based on the user's actual ratings ($o \succ_u j$). The mean of their predicted preference (Equation 11), the percentage of predictions that match with actual ratings (Equation 12), and the median of per user percentage of match are reported. When evaluating a recommendation, the percentages of random hold out items that are ranked higher than 20 (out of 100) using two different ranking method (M1 and M2) are shown. See Table 3 in Appendix for details about each of the comparing models.

| Dataset | Model | Consistency | | | Recommendation | |
|---|---|---|---|---|---|---|
| | | mean $x_{uij}$ | match (%) | per user (%) | M1 (%) | M2 (%) |
| Netflix | SAD | $\mathbf{0.024 \pm 0.012}$ | $\mathbf{33.7 \pm 0.5}$ | $18.7 \pm 0.3$ | $83.3 \pm 0.3$ | $83.9 \pm 0.3$ |
| | BPR | $-0.019 \pm 0.012$ | $32.6 \pm 0.5$ | $12.7 \pm 0.3$ | $81.8 \pm 0.4$ | $81.7 \pm 0.3$ |
| | SVD | $-0.824 \pm 0.032$ | $10.1 \pm 0.1$ | $0.0 \pm 0.0$ | $2.0 \pm 0.5$ | $2.1 \pm 0.3$ |
| | MF | $-0.018 \pm 0.058$ | $8.1 \pm 0.5$ | $0.0 \pm 0.0$ | $74.8 \pm 2.6$ | $45.4 \pm 8.7$ |
| | PMF | $0.003 \pm 0.015$ | $33.1 \pm 0.3$ | $\mathbf{26.9 \pm 0.3}$ | $21.1 \pm 0.3$ | $20.2 \pm 0.3$ |
| | FM | $-0.538 \pm 0.316$ | $12.9 \pm 0.2$ | $8.4 \pm 0.1$ | $\mathbf{86.4 \pm 0.2}$ | $81.6 \pm 0.3$ |
| | NCF | $-0.025 \pm 0.043$ | $13.1 \pm 4.4$ | $5.6 \pm 1.2$ | $86.0 \pm 0.3$ | $\mathbf{85.9 \pm 2.2}$ |
| | $\beta$-VAE | $-0.011 \pm 0.017$ | $21.1 \pm 0.9$ | $9.7 \pm 2.1$ | $35.7 \pm 3.1$ | $31.0 \pm 5.7$ |
| Movie -Lens | SAD | $\mathbf{0.120 \pm 0.014}$ | $\mathbf{36.4 \pm 0.6}$ | $\mathbf{29.9 \pm 0.7}$ | $82.9 \pm 0.4$ | $82.1 \pm 0.4$ |
| | BPR | $0.083 \pm 0.012$ | $35.7 \pm 0.6$ | $25.2 \pm 0.6$ | $77.7 \pm 0.5$ | $76.9 \pm 0.4$ |
| | SVD | $-0.324 \pm 0.011$ | $7.0 \pm 0.4$ | $0.0 \pm 0.0$ | $3.5 \pm 0.1$ | $3.1 \pm 0.2$ |
| | MF | $0.024 \pm 0.103$ | $19.0 \pm 0.5$ | $0.0 \pm 0.0$ | $47.8 \pm 1.0$ | $27.0 \pm 0.7$ |
| | PMF | $0.027 \pm 0.016$ | $32.7 \pm 0.4$ | $26.4 \pm 0.4$ | $27.3 \pm 0.8$ | $22.8 \pm 0.5$ |
| | FM | $0.103 \pm 0.025$ | $21.7 \pm 0.3$ | $18.1 \pm 0.3$ | $78.8 \pm 0.3$ | $76.3 \pm 0.4$ |
| | NCF | $-0.241 \pm 0.346$ | $24.0 \pm 1.9$ | $14.5 \pm 2.0$ | $\mathbf{90.4 \pm 1.5}$ | $\mathbf{90.1 \pm 2.1}$ |
| | $\beta$-VAE | $-0.120 \pm 0.301$ | $13.2 \pm 2.1$ | $10.9 \pm 2.4$ | $71.0 \pm 0.3$ | $69.9 \pm 0.7$ |
| Food -Com | SAD | $\mathbf{-0.329 \pm 0.002}$ | $\mathbf{14.9 \pm 0.5}$ | $0.0 \pm 0.0$ | $24.7 \pm 0.2$ | $23.9 \pm 0.2$ |
| | BPR | $-1.276 \pm 0.009$ | $5.9 \pm 0.5$ | $0.0 \pm 0.0$ | $23.2 \pm 0.3$ | $21.3 \pm 0.3$ |
| | SVD | $-5.152 \pm 0.039$ | $1.1 \pm 0.1$ | $0.0 \pm 0.0$ | $0.0 \pm 0.0$ | $0.0 \pm 0.0$ |
| | MF | $-1.956 \pm 0.161$ | $6.4 \pm 2.0$ | $0.0 \pm 0.0$ | $27.8 \pm 5.5$ | $24.6 \pm 6.1$ |
| | PMF | $-0.434 \pm 0.031$ | $4.1 \pm 3.5$ | $0.0 \pm 0.0$ | $20.2 \pm 0.6$ | $21.0 \pm 1.1$ |
| | FM | $-5.324 \pm 0.247$ | $9.3 \pm 1.7$ | $0.0 \pm 0.0$ | $35.9 \pm 0.2$ | $35.1 \pm 0.3$ |
| | NCF | $-11.362 \pm 0.852$ | $8.9 \pm 0.7$ | $0.0 \pm 0.0$ | $\mathbf{38.2 \pm 4.8}$ | $\mathbf{37.1 \pm 5.5}$ |
| | $\beta$-VAE | $-3.127 \pm 0.426$ | $10.2 \pm 1.1$ | $0.0 \pm 0.0$ | $16.3 \pm 3.1$ | $16.1 \pm 3.7$ |

To evaluate models' performance in recommendation, we create an item set containing 100 non-interacted items by randomly sampling for each user. We combine the hold out item with the 100 items to form a test set, and examine whether models are able to rank the hold out item high in the test set (Hit Ratio (He et al., 2017)). SAD faces some unconventional challenges (hence opportunities) in producing a ranking when violation of transitivity exists. Consider three items $i$, $j$ and $t$. With transitivity, if user has $i \succ j$ and $j \succ t$, then $i \succ t$ must hold. However, SAD can result in a scenario in which $t \succ i$, forming a preference cycle among the three items, in which case no ranking can be inferred. We propose two methods using pairwise comparisons for SAD in the evaluation. In the first method (M1), the number of non-interacted items in the test set that are more preferrable than the hold out item is dubbed as its rank. In a second method (M2), we use the ratings from interacted items kept in training set and calculate the proportion of interacted items that are less preferable than a test item. The proportion is used as a score to rank items in the test set. We calculate the percentage of hold out items that are ranked higher than 20 in all the hold out items. SAD is among the top three best models that rank the hold out items in top 20 (Table 1). We argue that since our model's predictions match better with user's actual ratings, it is able to bring additional information by introducing diversity of items into recommendation, while respecting users's potential preferences. In Appendix, we illustrate examples in which SAD produces model predictions consistent with true ratings while other SOTA models fail.

## 7 Discussions

We proposed a new tensor decomposition model for collaborative filtering with implicit feedback. In contrast to traditional models, we introduced a new set of non-negative latent vectors for items. While respecting anti-symmetricity of parameters, the new vectors generalized the standard dot products for calculating user-item preferences to general inner products, allowing items to interact when evaluating their relative preferences. When such vectors were all set to 1's, our model reduced to standard collaborative filtering models. We allowed their values to be different from 1, enabling the violation of the known transitivity property. The proposed model generated accurate recommendations across multiple real world datasets examined.

The existence of violation of transitivity has profound implication in real world applications. The items that violate the property form cycles in the directed graph implied by pairwise preferences of a user. For example, consider three items $i$, $j$ and $t$. Transitivity implies if user $u$ has $i \succ j$ and $j \succ t$, then $t \succ i$ must hold. Violation of transitivity suggests $t \succ i$, forming a preference cycle among the three items. When SAD detects such violations, even a strong prior, $l_1$ regularization, suggesting otherwise, it indicates that users themselves may not have a linear mental model in terms of ranking items. When making recommendations, the items forming a cycle can be selected as a group, and users who are not certain which one in the group is the most relevant can be exposed to the entire group, increasing the possibility of producing a hit.

When evaluating consistency in Section 6, we used users' actual ratings to determine pairwise preference between items. Many items in the datasets however had collided ratings due to the limited values those ratings can take. We chose the unevenly rated items with partial orders as ground truth during evaluation. The partial orders are a proxy for users' true preferences, and using them is the best we can do to evaluate whether our model is able to produce consistent predictions. For evenly rated items, there is no ground truth available to allow us to evaluate the performance.

This topic touches one fundamental aspect in our existing rating system - using an integer valued score with limited range as a sole reflection of user's preference. Traditional collaborative filtering models pander to this system, by assuming there is a linear ranking among items, and the ranking is dictated by a real number representing the preference strength. However, reality can hold evidence that violates the assumption. For example, Mazurek & Perzina (2017) demonstrated only a small proportion of real world pairwise comparisons satisfy complete transitivity. Li et al. (2013) showed the ubiquity of a user providing very different rating scores on closely correlated items, producing self-contradictions. In addition to noisiness in user ratings, the observed self-contradictions is a manifestation of the collision between user's mental model and the rating system. SAD innovates by providing a novel methodology to mathematically parameterize this mental model.

There can be parameterizations that both respect anti-symmetric property of $X_{u::}$, and at the same time allow the violation of transitivity property other than Equation 8. For example, instead of introducing the new vector $\tau_i \in \mathbb{R}_+^k$ for every item, one can model $x_{uij}$ as

$$x_{uij} = \sum_{h=1}^{k} \xi_{uh}(\eta_{hi} - \eta_{hj})\tau_{hij},$$

with $T_{h::} = \{\tau_{hij}\}_{1 \le i,j \le m} \in \mathbb{R}^{m \times m}$. $T_{h::}$ in this parameterization can be viewed as an item interaction matrix for $h$-th factor. Additional structures can be assigned to $T_{h::}$ to enable parsimoniousness, such as letting $T_{h::}$ be symmetric (in order to make $X_{u::}$ anti-symmetric) and modeling it as a low rank matrix with $\tau_{hij} = \langle \alpha_{hi}, \beta_{hj} \rangle$, and $\alpha_{hi}, \beta_{hj} \in \mathbb{R}^\kappa$. In our current work we focused on an efficient model with a simple interpretation. We delegate the research of exploring alternative parameterizations to future work.

In this paper, we restrict our scope to consider collaborative filterings for implicit feedbacks. SAD can be applied to datasets beyond the scope. For instances, datasets with explicit ratings contain partial orders that can be leveraged directly during model fitting, instead of being used to evaluate model consistency in a post-hoc manner as in current work. Such datasets include the ones considered in Section 6, and numerous others such as Yandex challenge and KDD Cup 2012. Other datasets that contain actual pairwise comparisons such as the one created by Pavlichenko & Ustalov (2021) are a natural fit to SAD as well. We expect the power of SAD can be further enhanced with neural network components integrated.

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

## A Properties of Real World Datasets

Table 2: Properties of the three real word datasets used in Section 6

| Dataset | #Users | #Items | #Ratings | Sparsity | Quantiles of #Ratings/User (min/5%/50%/95%/max) |
|---------|--------|--------|----------|----------|-------------------------------------------------|
| Netflix | $10,000$ | $8,693$ | $1,044,318$ | 98.80% | $(1/6/46/400/8,237)$ |
| Movie-Lens | $6,040$ | $3,706$ | $1,000,209$ | 95.53% | $(20/23/96/556/2,314)$ |
| Food-Com | $17,482$ | $1,358$ | $145,431$ | 99.39% | $(1/1/4/30/878)$ |

## B Gibbs Sampler for Posterior Inference of SAD

We derive an efficient Gibbs sampling algorithm as a complement to the SGD algorithm in the main paper. The Gibbs sampling algorithm has the advantage of producing accurate uncertainty estimation of unknowns under Bayesian inference by drawing parameter samples from the posterior distribution. The algorithm is an application of Bayesian Probit regression to the current setting. Specifically, we replace the original logistic parameterization in Equation 1 with the following equation:

$$p(d_{uij} = 1 | \Theta) := \Phi(x_{uij}), \tag{13}$$

where $\Phi(x_{uij})$ is the CDF of a Gaussian distribution with mean $x_{uij}$ and variance 1. By augmenting the model with a hidden tensor $Z = \{z_{uij}\}$, where $z_{uij} = x_{uij} + \epsilon_{uij}$ and $\epsilon_{uij} \overset{\text{i.i.d}}{\sim} N(0,1)$, the Probit model is equivalent to

$$d_{uij} = \begin{cases} 1 & z_{uij} > 0 \\ -1 & z_{uij} \leq 0 \end{cases}.$$

With this new model, an efficient Gibbs sampling algorithm can be derived. As a toy example, we assign spherical Gaussian priors to rows of $\Xi$, $H$ and $T$ independently. With the likelihood defined in Equation 13, the following conditional posterior distributions can be derived.

**Posterior of $z_{uij}$**

$$z_{uij} | \Xi, H, T \sim \begin{cases} N_+(x_{uij}, 1) & \text{if } d_{uij} = 1 \\ N_-(x_{uij}, 1) & \text{if } d_{uij} = -1 \end{cases},$$

where $N_+(\mu, \sigma)$ and $N_-(\mu, \sigma)$ are truncated Gaussian distributions on positive and negative quadrants respectively.

**Posterior of $\xi_u$ with $u = 1, \cdots, n$**

$$\xi_u | Z, \Xi \setminus \xi_u, H, T \sim N_k(\Sigma_u^{\xi} (\Psi_u^{\xi})^{\top} \bar{z}_u^{\xi}, \Sigma_u^{\xi}),$$

where $(\Sigma_u^{\xi})^{-1} = (\Psi_u^{\xi})^{\top} (\Psi_u^{\xi}) + I,$

$$\Psi_u^{\xi} = \begin{pmatrix} \eta_{11}\tau_{12} - \eta_{12}\tau_{11} & \eta_{21}\tau_{22} - \eta_{22}\tau_{21} & \cdots & \eta_{k1}\tau_{k2} - \eta_{k2}\tau_{k1} \\ \eta_{11}\tau_{13} - \eta_{13}\tau_{11} & \eta_{21}\tau_{23} - \eta_{23}\tau_{21} & \cdots & \eta_{k1}\tau_{k3} - \eta_{k3}\tau_{k1} \\ \vdots & \vdots & \ddots & \vdots \\ \eta_{12}\tau_{13} - \eta_{13}\tau_{12} & \eta_{22}\tau_{23} - \eta_{23}\tau_{22} & \cdots & \eta_{k2}\tau_{k3} - \eta_{k3}\tau_{k2} \\ \vdots & \vdots & \ddots & \vdots \\ \eta_{1,m-1}\tau_{1m} - \eta_{1m}\tau_{1,m-1} & \eta_{2,m-1}\tau_{2m} - \eta_{2m}\tau_{2,m-1} & \cdots & \eta_{k,m-1}\tau_{km} - \eta_{km}\tau_{k,m-1} \end{pmatrix}$$
$$\in \mathbb{R}^{m(m-1)/2 \times k},$$

and $\bar{z}_u^{\xi} = [z_{u12}, z_{u13}, \cdots, z_{u23}, z_{u24}, \cdots, z_{u,m-1,m}]^{\top} \in \mathbb{R}^{m(m-1)/2}.$

**Posterior of $\eta_i$ with $i = 1, \cdots, m$**

$$\eta_i | Z, \Xi, H \setminus \eta_i, T \sim N_k(\Sigma_i^\eta (\Psi_i^\eta)^\top \bar{z}_i^\eta, \Sigma_i^\eta),$$

where $(\Sigma_i^\eta)^{-1} = (\Psi_i^\eta)^\top (\Psi_i^\eta) + I$,

$$
\Psi_i^\eta = \begin{pmatrix}
\xi_{11}\tau_{11} & \xi_{21}\tau_{21} & \cdots & \xi_{k1}\tau_{k1} \\
\xi_{11}\tau_{12} & \xi_{21}\tau_{22} & \cdots & \xi_{k1}\tau_{k2} \\
\vdots & \vdots & \ddots & \vdots \\
\xi_{11}\tau_{1,i-1} & \xi_{21}\tau_{2,i-1} & \cdots & \xi_{k1}\tau_{k,i-1} \\
\xi_{11}\tau_{1,i+1} & \xi_{21}\tau_{2,i+1} & \cdots & \xi_{k1}\tau_{k,i+1} \\
\vdots & \vdots & \ddots & \vdots \\
\xi_{11}\tau_{1m} & \xi_{21}\tau_{2m} & \cdots & \xi_{k1}\tau_{km} \\
\xi_{12}\tau_{11} & \xi_{22}\tau_{21} & \cdots & \xi_{k2}\tau_{k1} \\
\vdots & \vdots & \ddots & \vdots \\
\xi_{12}\tau_{1,i-1} & \xi_{22}\tau_{2,i-1} & \cdots & \xi_{k2}\tau_{k,i-1} \\
\xi_{12}\tau_{1,i+1} & \xi_{22}\tau_{2,i+1} & \cdots & \xi_{k2}\tau_{k,i+1} \\
\vdots & \vdots & \ddots & \vdots \\
\xi_{12}\tau_{1m} & \xi_{22}\tau_{2m} & \cdots & \xi_{k2}\tau_{km} \\
\vdots & \vdots & \ddots & \vdots \\
\xi_{1n}\tau_{11} & \xi_{2n}\tau_{21} & \cdots & \xi_{kn}\tau_{k1} \\
\vdots & \vdots & \ddots & \vdots \\
\xi_{1n}\tau_{1,i-1} & \xi_{2n}\tau_{2,i-1} & \cdots & \xi_{kn}\tau_{k,i-1} \\
\xi_{1n}\tau_{1,i+1} & \xi_{2n}\tau_{2,i+1} & \cdots & \xi_{kn}\tau_{k,i+1} \\
\vdots & \vdots & \ddots & \vdots \\
\xi_{1n}\tau_{1m} & \xi_{2n}\tau_{2m} & \cdots & \xi_{kn}\tau_{km}
\end{pmatrix} \in \mathbb{R}^{n(m-1) \times k},
$$

$$
\bar{z}_i^\eta = \begin{pmatrix}
z_{1i1} + \sum_{h=1}^{k} \xi_{h1}\tau_{hi}\eta_{h1} \\
z_{1i2} + \sum_{h=1}^{k} \xi_{h1}\tau_{hi}\eta_{h2} \\
\vdots \\
z_{1,i,i-1} + \sum_{h=1}^{k} \xi_{h1}\tau_{hi}\eta_{h,i-1} \\
z_{1,i,i+1} + \sum_{h=1}^{k} \xi_{h1}\tau_{hi}\eta_{h,i+1} \\
\vdots \\
z_{1im} + \sum_{h=1}^{k} \xi_{h1}\tau_{hi}\eta_{h,m} \\
z_{2i1} + \sum_{h=1}^{k} \xi_{h2}\tau_{hi}\eta_{h1} \\
\vdots \\
z_{2,i,i-1} + \sum_{h=1}^{k} \xi_{h2}\tau_{hi}\eta_{h,i-1} \\
z_{2,i,i+1} + \sum_{h=1}^{k} \xi_{h2}\tau_{hi}\eta_{h,i+1} \\
\vdots \\
z_{2im} + \sum_{h=1}^{k} \xi_{h2}\tau_{hi}\eta_{hm} \\
\vdots \\
z_{ni1} + \sum_{h=1}^{k} \xi_{hn}\tau_{hi}\eta_{h1} \\
\vdots \\
z_{n,i,i-1} + \sum_{h=1}^{k} \xi_{hn}\tau_{hi}\eta_{h,i-1} \\
z_{n,i,i+1} + \sum_{h=1}^{k} \xi_{hn}\tau_{hi}\eta_{h,i+1} \\
\vdots \\
z_{nim} + \sum_{h=1}^{k} \xi_{hn}\tau_{hi}\eta_{hm}
\end{pmatrix} \in \mathbb{R}^{n(m-1)}.
$$

In the above notation, we take advantage of the anti-symmetric property of $Z_{u::}$ and assume $z_{uij} = -z_{uji}$ when $i > j$.

**Posterior of $\tau_j$ with $j = 1, \cdots, m$**

$$\tau_j | Z, \Xi, H, T \setminus \tau_j \sim N_k^+ (\Sigma_j^\tau (\Psi_j^\tau)^\top \bar{z}_j^\tau, \Sigma_j^\tau),$$

where $(\Sigma_j^\tau)^{-1} = (\Psi_j^\tau)^\top (\Psi_j^\tau) + I$,

$$\Psi_j^\tau = \begin{pmatrix}
\xi_{11}\eta_{11} & \xi_{21}\eta_{21} & \cdots & \xi_{k1}\eta_{k1} \\
\xi_{11}\eta_{12} & \xi_{21}\eta_{22} & \cdots & \xi_{k1}\eta_{k2} \\
\vdots & \vdots & \ddots & \vdots \\
\xi_{11}\eta_{1,j-1} & \xi_{21}\eta_{2,j-1} & \cdots & \xi_{k1}\eta_{k,j-1} \\
\xi_{11}\eta_{1,j+1} & \xi_{21}\eta_{2,j+1} & \cdots & \xi_{k1}\eta_{k,j+1} \\
\vdots & \vdots & \ddots & \vdots \\
\xi_{11}\eta_{1m} & \xi_{21}\eta_{2m} & \cdots & \xi_{k1}\eta_{km} \\
\xi_{12}\eta_{11} & \xi_{22}\eta_{21} & \cdots & \xi_{k2}\eta_{k1} \\
\vdots & \vdots & \ddots & \vdots \\
\xi_{12}\eta_{1,j-1} & \xi_{22}\eta_{2,j-1} & \cdots & \xi_{k2}\eta_{k,j-1} \\
\xi_{12}\eta_{1,j+1} & \xi_{22}\eta_{2,j+1} & \cdots & \xi_{k2}\eta_{k,j+1} \\
\vdots & \vdots & \ddots & \vdots \\
\xi_{12}\eta_{1m} & \xi_{22}\eta_{2m} & \cdots & \xi_{k2}\eta_{km} \\
\vdots & \vdots & \ddots & \vdots \\
\xi_{1n}\eta_{11} & \xi_{2n}\eta_{21} & \cdots & \xi_{kn}\eta_{k1} \\
\vdots & \vdots & \ddots & \vdots \\
\xi_{1n}\eta_{1,j-1} & \xi_{2n}\eta_{2,j-1} & \cdots & \xi_{kn}\eta_{k,j-1} \\
\xi_{1n}\eta_{1,j+1} & \xi_{2n}\eta_{2,j+1} & \cdots & \xi_{kn}\eta_{k,j+1} \\
\vdots & \vdots & \ddots & \vdots \\
\xi_{1n}\eta_{1m} & \xi_{2n}\eta_{2m} & \cdots & \xi_{kn}\eta_{km}
\end{pmatrix} \in \mathbb{R}^{n(m-1) \times k},$$

$$\bar{z}_j^\tau = \begin{pmatrix}
-z_{1j1} + \sum_{h=1}^k \xi_{h1}\tau_{h1}\eta_{hj} \\
-z_{1j2} + \sum_{h=1}^k \xi_{h1}\tau_{h2}\eta_{hj} \\
\vdots \\
-z_{1j,j-1} + \sum_{h=1}^k \xi_{h1}\tau_{h,j-1}\eta_{hj} \\
-z_{1j,j+1} + \sum_{h=1}^k \xi_{h1}\tau_{h,j+1}\eta_{hj} \\
\vdots \\
-z_{1jm} + \sum_{h=1}^k \xi_{h1}\tau_{hm}\eta_{hj} \\
-z_{2j1} + \sum_{h=1}^k \xi_{h2}\tau_{h1}\eta_{hj} \\
\vdots \\
-z_{2j,j-1} + \sum_{h=1}^k \xi_{h2}\tau_{h,j-1}\eta_{hj} \\
-z_{2j,j+1} + \sum_{h=1}^k \xi_{h2}\tau_{h,j+1}\eta_{hj} \\
\vdots \\
-z_{2jm} + \sum_{h=1}^k \xi_{h2}\tau_{hm}\eta_{hj} \\
\vdots \\
-z_{n,j,j-1} + \sum_{h=1}^k \xi_{hn}\tau_{h,j-1}\eta_{hj} \\
-z_{n,j,j+1} + \sum_{h=1}^k \xi_{hn}\tau_{h,j+1}\eta_{hj} \\
\vdots \\
-z_{njm} + \sum_{h=1}^k \xi_{hn}\tau_{hm}\eta_{hj}
\end{pmatrix} \in \mathbb{R}^{n(m-1)}.$$

Similar to $\bar{z}_i^\eta$, we take advantage of the anti-symmetric property of $Z_{u::}$ and assume $z_{uij} = -z_{uji}$ when $i > j$.

## C   Methods Considered in the Real World Application

Table 3: Specifics & hyperparameters for models used when applying to real world datasets.

| Model | Parameter | Values |
|---|---|---|
| SAD | Implementation | Supplementary Code |
| | Learning Rate | $[0.001, 0.002, 0.005, 0.01, 0.02, 0.05, 0.1]$ |
| | # Epochs | $[2, 5, 10, 20, 50]$ |
| | $l_2$ Reg | $[0.05, 0.01, 0.005, 0.001]$ |
| | $l_1$ Reg | $0.01$ |
| BPR (Rendle et al., 2009) | Implementation | Supplementary Code |
| | Learning Rate | $[0.001, 0.002, 0.005, 0.01, 0.02, 0.05, 0.1]$ |
| | # Epochs | $[2, 5, 10, 20, 50]$ |
| | $l_2$ Reg | $[0.05, 0.01, 0.005, 0.001]$ |
| SVD | Implementation | Surprise (package) (Hug, 2020) |
| | Learning Rate | $[0.001, 0.002, 0.005, 0.01, 0.02, 0.05, 0.1]$ |
| | # Epochs | $[2, 5, 10, 20, 50]$ |
| | Regularization | $[0.05, 0.01, 0.005, 0.001]$ |
| Matrix Factorization (MF) | Implementation | Cornac (package) (Salah et al., 2020) |
| | Learning Rate | $[0.001, 0.002, 0.005, 0.01, 0.02, 0.05, 0.1]$ |
| | # Epochs | $[2, 5, 10, 20, 50]$ |
| | $\lambda$ Reg | $[0.05, 0.01, 0.005, 0.001]$ |
| Probabilistic Matrix Factorization (PMF) (Mnih & Salakhutdinov, 2008) | Implementation | Cornac (package) (Salah et al., 2020) |
| | Learning Rate | $[0.001, 0.002, 0.005, 0.01, 0.02, 0.05, 0.1]$ |
| | # Epochs | $[2, 5, 10, 20, 50]$ |
| | $\lambda$ Reg | $[0.05, 0.01, 0.005, 0.001]$ |
| Factorization Machine (FM) (Rendle, 2010) | Implementation | RankFM (package) |
| | Learning Rate | $[0.001, 0.002, 0.005, 0.01, 0.02, 0.05, 0.1]$ |
| | # Epochs | $[2, 5, 10, 20, 50]$ |
| | $l_2$ Reg | $[0.05, 0.01, 0.005, 0.001]$ |
| Neural Collaborative Filtering (NCF) (He et al., 2017) | Implementation | MSFT recommenders (package) (Graham et al., 2019) |
| | Learning Rate | $[0.001, 0.002, 0.005, 0.01, 0.02, 0.05, 0.1]$ |
| | # Epochs | $[2, 5, 10, 20, 50]$ |
| | Batch size | $[128, 256, 512, 1024]$ |
| | Network | Three layers MLP with sizes $[128, 64, 32]$ |
| Variational AutoEncoder ($\beta$-VAE) (Liang et al., 2018) | Implementation | MSFT recommenders (package) (Graham et al., 2019) |
| | $\beta$ parameter | $[0.001, 0.002, 0.005, 0.01, 0.02, 0.05, 0.1]$ |
| | # Epochs | $[2, 5, 10, 20, 50]$ |
| | Batch size | $[128, 256, 512, 1024]$ |

## D   Real World Examples

During LOO evaluation, true preferences between a hold out item and the rest user interacted items in training set are determined based on users' actual ratings. Cases in which our model's predictions are consistent with true preferences while alternative models are not are shown in Table 4. We select 10 such cases from each of the three real world datasets. Model predictions together with item descriptions are shown in the table.

Table 4: Examples where SAD produces a consistent prediction ($x_{uij} > 0$ & $p_{uij} > 0.5$) while BPR fails ($x_{uij} < 0$ & $p_{uij} < 0.5$).

| Dataset | $u$-th user | $i$-th item (rating) | $j$-th item (rating) | $x_{uij} \mid p_{uij}$ SAD | BPR |
|---|---|---|---|---|---|
| Netflix | '1381599' | Last of the Dogmen (5) | Look at Me (2) | 0.35 \| 0.59 | $-0.22$ \| 0.44 |
| | '1243460' | Joe Kidd (5) | Scenes of the Crime (1) | 0.80 \| 0.69 | $-0.27$ \| 0.43 |
| | '581011' | The Professional (5) | The Bourne Identity (2) | 0.80 \| 0.69 | $-0.32$ \| 0.42 |
| | '632823' | Lara Croft: Tomb Raider: The Cradle of Life (4) | The Cookout (2) | 0.27 \| 0.57 | $-1.36$ \| 0.20 |
| | '429299' | The Worst Witch (4) | A Chorus Line (1) | 0.85 \| 0.70 | $-0.76$ \| 0.32 |
| | '127356' | Trading Spaces: Great Kitchen Designs and More! (4) | White Oleander (2) | 0.30 \| 0.57 | $-1.24$ \| 0.22 |
| | '2264661' | Free Tibet (5) | Native American Medicine (3) | 1.25 \| 0.78 | $-0.05$ \| 0.49 |
| | '581011' | The Dream Catcher (4) | The Bourne Identity (2) | 0.90 \| 0.71 | $-0.37$ \| 0.41 |
| | '1368371' | Zombie Holocaust (4) | Gothika (1) | 0.71 \| 0.67 | $-0.31$ \| 0.42 |
| | '1243460' | Moog (3) | Scenes of the Crime (1) | 0.98 \| 0.73 | $-1.34$ \| 0.21 |
| Movie-Lens | '1318' | High Fidelity (5) | Jimmy Hollywood (3) | 0.85 \| 0.70 | $-1.26$ \| 0.22 |
| | '1250' | American Beauty (5) | eXistenZy (3) | 1.44 \| 0.81 | $-0.14$ \| 0.47 |
| | '4166' | My Fair Lady (5) | Problem Child 2 (1) | 0.51 \| 0.63 | $-0.78$ \| 0.31 |
| | '153' | American Beauty (4) | Spice World (1) | 1.07 \| 0.75 | $-0.08$ \| 0.48 |
| | '2160' | Harold and Maude (4) | The Brady Bunch Movie (1) | 0.34 \| 0.58 | $-0.73$ \| 0.32 |
| | '4692' | Blade Runner (5) | The Newton Boys (3) | 0.65 \| 0.66 | $-0.38$ \| 0.41 |
| | '2385' | Braveheart (4) | Voyage to the Bottom of the Sea (3) | 0.98 \| 0.73 | $-0.56$ \| 0.36 |
| | '4756' | Galaxy Quest (3) | Felicia's Journey (2) | 1.01 \| 0.73 | $-0.21$ \| 0.45 |
| | '4439' | The Maltese Falcon (4) | Entrapment (3) | 0.45 \| 0.61 | $-0.57$ \| 0.36 |
| | '3021' | L.A. Confidential (5) | Fatal Attraction (3) | 0.64 \| 0.66 | $-0.37$ \| 0.41 |
| Food-Com | '148323' | Best Ever Banana Cake \w Cream Cheese Frosting (5) | Crock Pot Garlic Brown Sugar Chicken (0) | 0.16 \| 0.54 | $-3.51$ \| 0.03 |
| | '424008' | Glazed Cinnamon Rolls, Bread Machine (5) | Japanese Mum's Chicken (0) | 0.31 \| 0.57 | $-0.38$ \| 0.41 |
| | '428423' | Crock Pot Stifado (5) | Best Ever and Most Versatile Muffins (3) | 0.30 \| 0.57 | $-2.99$ \| 0.05 |
| | '733257' | Banana Banana Bread (2) | Low Fat Oatmeal Muffins (0) | 0.32 \| 0.58 | $-2.31$ \| 0.09 |
| | '340980' | Funky Chicken \w Sesame Noodles (3) | Amanda's Thai Peanut (1) | 0.56 \| 0.64 | $-0.77$ \| 0.32 |
| | '573772' | Delicious Chicken Pot Pie (5) | Amish Oven Fried Chicken (1) | 1.28 \| 0.78 | $-1.21$ \| 0.23 |
| | '1477540' | Cinnabon Cinnamon Rolls by Todd Wilbur (4) | Amanda's Cheese Pound Cake (0) | 0.81 \| 0.69 | $-2.59$ \| 0.07 |
| | '268644' | Baked Tilapia \w Lots of Spice (5) | Southern Fried Salmon Patties (2) | 0.38 \| 0.59 | $-3.35$ \| 0.03 |
| | '762742' | Easy Peezy Pizza Dough & Bread Machine Pizza Doug (5) | Fresh Orange Muffins (1) | 0.98 \| 0.73 | $-2.31$ \| 0.09 |
| | '212558' | Steak or Chicken Fajitas (5) | Thai Style Ground Beef (3) | 1.86 \| 0.87 | $-0.11$ \| 0.47 |

