# OpenReview forum: "Consistent Collaborative Filtering via Tensor Decomposition"
_TMLR — Accepted by TMLR_

### Review · Reviewer_aiYy · 2023-05-01

**Summary Of Contributions:**

The authors developed a new collaborative filtering algorithm, Sliced Anti-symmetric Decomposition (SAD), for recommender systems. This new algorithm extended Bayesian Personalized Ranking (BPR) by introducing one additional latent vector to each item. Collapsing this additional latent vector to 1 will result in the standard BPR model so this new algorithm can be considered as a generalization of BPR and matrix factorization. Allowing it to have values other than 1 can model users' nonlinear mental models when evaluating items by violating of the transitivity property that is followed by most existing collaborative filtering algorithms.

**Audience:**

Yes

**Broader Impact Concerns:**

None.

**Claims And Evidence:**

Yes

**Requested Changes:**

1. I would suggest the authors to further push their proposed approach to be more innovative. Combining the idea in this paper with what they mentioned in their future work, integration with neural network components, may be a good direction to explore.
2. Some typos need to be fixed, such as
    * "we show that SAD is not only able to more" in Abstract.
    * "Some details of the three datasets can be found in. " in Section 6.
    * missing caption of Figure 3.


**Strengths And Weaknesses:**

Strengths
1. This paper studies the transitivity property followed by most existing collaborative algorithms. It's an interesting topic that has been overlooked. The argument is that people's mental model of evaluating items does not follow this property is valid and well explained. It's intuitive that cycles in pairwise comparisons may happen a lot in reality. It's important to model this phenomenon.
2. The authors conducted experiments over multiple datasets with reasonable size. They also compared their approaches with multiple baselines.
3. Most content of this paper is well-explained and easy to follow.
4. The M1 and M2 metrics used in evaluations use pairwise comparison to rank items. It's nice to see the authors develop them as their proposed model relies on pairwise comparison while most evaluation metrics, which are developed for listwise ranking, may not be the best to evaluate their models.

Weaknesses
1. While the problem studied in this paper is interesting, the proposed solution is not very impressive.
    * First of all, compared with most recent advancements in recommender systems, the proposed approach is very simple and incremental.
    * Secondly, if we stick with the approach of adding one additional matrix to BPR, the proposed solution of modeling the third matrix as another item matrix may not be optimal. One alternative is to model the third matrix as an interaction matrix between item $i$ and item $j$, so $x_{uij} = \sum_{h=1}^{k}\xi_{hu}(\eta_{hi} - \eta_{hj})\tau_{hij}$. $\tau_{hij}$ can be further modeled as the inner product of two vectors, one from item $i$ and one from item $j$. This modeling approach is more intuitive to me than the one proposed in this paper as it assumes that which aspect (latent dimension) the user care about is determined by some interactions between the two items.
2. While it's a strength that the authors use multiple baselines for comparison, it would be better if they can consider more variations of BPR and matrix factorization/collaborative filtering. Researchers have developed tons of algorithms to extend them and some have considered introducting additional matrices to it, such as https://arxiv.org/pdf/1711.08379.pdf.

---

> ### Author Response · Authors · 2023-05-19
> **To reviewer aiYy**
>
> > Strengths
>
> Thanks for the summarization. See below for our responses to the rest of the comments.
>
> > W1 a). First of all, ... incremental
>
> We believe the simplicity actually reflects the significance of our work. Simple solutions to complex problems often require the most advanced thinking and innovation. Our work challenges the fundamental assumption in most of current recommender systems - the existence of a linear ranking among items that is implied by the transitivity property. On the contrary, we let the data speak. When there are strong signals in the data indicating violation of transitivity, our model is able to identify them. In contrast, almost all of current recommender systems have fundamentally ruled out this possibility, regardless of whether there are evidences in the observed data. The simple mathematical solution displayed in our work demonstrates non-trivial advancement in current recommender systems. To recap, we propose a simple solution to a very important problem, and simplicity should be extra credit. We hope the reviewer agrees with us.
>
> > W1 b). Secondly, ... two items.
>
> Thanks for sharing this novel representation parameterization. With $\tau_{hij}$ being modeled as an inner product of two additional item vectors (e.g. $\alpha_{hi} \in \mathbb{R}^\kappa$, and $\beta_{hj} \in \mathbb{R}^\kappa$ with $\tau_{hij} =\langle \alpha_{hi}, \beta_{hj} \rangle$), we increase the number of parameters from the original $(n + 2m) \times k$ (parameters in $\Xi \in \mathbb{R}^{n \times k}$, $H \in \mathbb{R}^{m \times k}$ and $T \in \mathbb{R}^{m \times k}$ ) to $(n + m + 2 \kappa \cdot m) \times k$ , with the $2 \kappa \cdot m \cdot k$ parameters for $\alpha_{hi} \in \mathbb{R}^\kappa$ and $\beta_{hi} \in \mathbb{R}^\kappa$ for each latent dimension $h$ and every item $i$. Even with the simplest case in which $\kappa = 1$, the new parameterization still introduces $m \times k$ additional parameters compared with the model presented in our manuscript. In this work, we focused on the most basic parameterization that allows transitivity to be violated based on observed data. We leave the variations of the parameterization to future study. To clarify, we added the following paragraph to discussion in our revised manuscript.
>
>     There can be parameterizations that both respect anti-symmetric property of $X_{u::}$, and at the same time allow the violation of transitivity property other than the SAD model defined by Equation (8). In our current work we focused on an efficient model with a simple interpretation. We delegate the research of exploring alternative parameterizations to future work. We expect the power of SAD can be further enhanced with neural network components integrated.
>
> > W2. While ... such as ref.
>
> Thanks for sharing the reference. We agreed that the literature around matrix factorization (MF)/collaborative filtering (CF) models is vast, and enumerating and benchmarking all the models is infeasible. However, to the best of our knowledge, none of the existing work allows violation of transitivity in pairwise comparison (including the reference provided in the comment). SAD provides a basic and simple framework to allow such exceptions. As a consequence, we choose a few most basic MF/CF models as baselines to benchmark their performances on a common ground.
>
> > RC1. I would suggest ... to explore.
>
> We agree that there are multiple directions that can be explored as extensions of current work. We scope the model presented in current paper to be the inception of a basic and simple model that breaks the stereotypical assumption in existing MF/CF literature. We defer the extensions of SAD such as exploring alternative parameterizations and integration with neural networks to future work. Please see previous comment about innovation.
>
> > RC2. Some typos ...
>
> The typos listed above are all get fixed in the revised manuscript. Thanks for pointing out.

---

> > ### Comment · Reviewer_aiYy · 2023-06-21
> > **Response to authors' comments**
> >
> > I appreciate the author's detailed explanations to my questions. Most of my comments have been addressed and the revised version looks good to me. Sorry for the delay of this comment!

---

### Review · Reviewer_Kngd · 2023-05-03

**Summary Of Contributions:**

The article is interested in recommender systems. It introduces an extension of the BPR model, where the scalar $x_{uij}$ governing the probability that user $u$ prefers item $i$ over item $j$ is rescaled by a new set of parameters $\tau_{hi}\tau_{hj}$. In the experiments on 3 datasets, the proposed model enjoys a better consistency in terms of pairwise ranking than 7 other state-of-the-art models, including BPR.



**Audience:**

Yes

**Claims And Evidence:**

No

**Requested Changes:**

* (Critical) Rewrite the narrative of the proposed approach based on equation (10).
* (critical) Remove the paragraph "While respecting the anti-symmetry [...]".
* (strengthening the work) Rethink the section "Simulation study".
* (critical) Clarify the section "Applications for real data".
* (strengthen the paper) Add comparison with implicit feedback literature


**Strengths And Weaknesses:**

My view on the proposed model requires to rewrite Equation (8) as $x_{uij} = \sum_h \xi_{hu}(\zeta_{hi} - \zeta_{hj})\tau_{hi}\tau_{hj},$
where $\zeta_{hi} = \frac{\eta_{hi}}{\tau_{hi}}$. Let me denote (10) this equation.


### Strengths

* **A new model that can simultaneously model the ranking of items and the (non-)observation of this ranking**. As in BPR, the term $\zeta_{hi} - \zeta_{hj}$
in equation (10) measures the extent to which item $i$ is preferred over item $j$ given feature $h$ ($\eta_{hi} > \eta_{hj}$), and the weighted sum of these terms (weighted only by $\xi_{hu}$) indicates whether item $i$ is preferred over item $j$ given user $u$. The addition of $\tau_{hi}\tau_{hj}$ allows the reduction of $x_{uij}$ when two items have a small probability of being present in the data. Indeed, when two items are present in the data, $d_{uij}$ is either -1 or 1 and can be fitted with $x_{uij}$ having a large value, while when one of the two items is not present in the data, even if there is a clear advantage for one of the two items, the observation is $d_{uij}=0$, forcing $x_{uij}$ to be close to 0.

### Weaknesses

* **Equation (8) is misleading and the narrative would be more accurate if based on Equation (10).**
* **The paragraph "While respecting the anti-symmetricity [...]" does not fit the current model.** In fact, in the current model, the term $x_{uij}$ is derived only from elements $i$ and $j$: other elements have no effect, the respective positions of the elements in the recommendation have no effect. To handle such a property, one has to look at models like in [1].
* **The contribution of the simulation study is limited.** Indeed, the parameters $n$, $m$, and $k$ are tiny compared to what can be expected from real data. Moreover, unlike the authors, I see BPR outperforming the proposed approach (SAD) in both settings when the percentage of missing information is high, and I cannot distinguish a leader when the percentage of missing information is low (while SAD should outperform BPR in setting (sim2)).
* **The contribution of the section "Applications for Real Data" is unclear.** First, the presentation of the criterion "consistency" is unclear, which forbids any conclusion from this criterion. Second, I do not see what information "mean $x_{uij}$" brings. Third, both consistency and recommendation seem to be related to the ranking of items, I do not understand what additional information the second criterion brings. Fourth, the use of non-interacted items to build a criterion is controversial because we do not know if they are non-interacted because the user is not interested in them or because he has not had the opportunity to consider them.
* **The paper should assess its relationship to the implicit feedback literature.** In my opinion, the main contribution of the paper is a way to "hide" the comparison between items when the comparison is not present in the data. This is typical of the implicit feedback literature. The paper should refer to this literature, compare to the state of the art article in the experimental part, and consider standard experimental protocols in the field.
* **Details**
	* Equation (5) is the same as the equation in section 4. I do not see its purpose
	* Why Figures 1, 2 and 3 are plotted much before they are referenced ?
	* Figure 3's caption is missing
	* The reference is missing in the sentence "Some details of the three datasets can be found in"


### Bibliography
[1] Lijing Qin, Shouyuan Chen, and Xiaoyan Zhu. Contextual Combinatorial Bandit
and its Application on Diversified Online Recommendation. SDM'14

---

> ### Author Response · Authors · 2023-05-19
> **To reviewer Kngd (to reduce characters we omitted the whole comments and only quote the first and last words)**
>
> > Strengths
> > A new model ... close to 0
>
> Thanks for suggesting to rewrite Eq (8) to the new form.
>
> 1. Multiple factors need to take into account when fitting $x_{uij}$. When entries in X are modeled independently, one can let $x_{uij} = \infty (-\infty)$ to fit $d_{uij} = 1 (-1)$. This scenario requires $nmm$ free parameters, and no information would be shared among users (or items) that have similar characteristics. With the assumption made in collaborative filtering models, $x_{uij}$ is modeled by a parsimonious equation $x_{uij} = \sum_h (\xi_{hu} \eta_{hi} - \xi_{hu} \eta_{hj})$, reducing the number of free parameters to $(n+m)k$. Available entries in D jointly impact estimate of $x_{uij}$.
>
> 2. When one of the two items is not present in the data, we are unable to infer the preference between the two. In this case, we actually exclude $d_{uij}$​ from evaluating our objective function (see Eq (9)). The value of $x_{uij}$ can be only estimated based on the impact from other observed entries, due to the coupling effects mentioned above. To reduce confusion, we removed the denotation of  $d_{uij} = 0$.
>
> We added a paragraph on Pg 2 (bottom) to clarify.
>
> > Equation (8) is ... Equation (10).
>
> We tend to prefer the parameterization in Eq (8) in the manuscript.
>
> 1. Eq (10) can be written as $\sum_h \xi_{hu}\zeta_{hi} \tau_{hi} \tau_{hj} - \xi_{hu}\zeta_{hj} \tau_{hi} \tau_{hj}$. The subscripts of $\zeta$ always coincide with one of $\tau_{hi} \tau_{hj}$, a sign of over parameterization. Letting $\eta_{hi} = \zeta_{hi} \tau_{hi}$ (and $\eta_{hj} = \zeta_{hj} \tau_{hj}$) simplifies the presentation.
>
> 2. The original parameterization is an inner product extension of original dot product. Without constraining $\tau_{hi}$ to positive $H$ and $T$ become unidentifiable - multiplying $-1$ to both $H$ and $T$ produces identical results. In fact it applies for any constant $c$ and $1/c$. Requiring $\tau_{hi}$ to be positive resolves the un-identifiability caused by sign flip; $l_1$ regularization at 1 discourages constant multiplication that moves $\tau_{hi}$ away from 1.
>
> We made clarification in a new paragraph above Section 4.4 on Pg 6.
>
> > The paragraph ... models like in [1].
>
> We apologize for the confusion. In fact, all available data in $D$ jointly impact parameter estimate of $x_{uij}$​. We have added a paragraph on Pg 2 to highlight it explicitly.
>
> The respective positions in the recommendation is a topic that is out of scope of current study in which we only consider static user item interactions. Agreed that to handle users' temporal behaviors, a sequence model (such as ref [1]) can be used.
>
> > The contribution ... (sim2)).
>
> The main goal of the simulation study is not to probe the performance under real world settings. Instead, we examine the model with the true parameters known ahead to understand the convergence and accuracy of our algorithm.
>
> In Sim2, with low level of data missing, SAD reached a significant lower error measured by Frobenius distance. We highlighted this in caption of Figure 4 in revision. With high level of missing, BPR achieved a lower error. We believe this is because the greater parsimoniousness of BPR. When significant amount of data are missing, complex models can struggle in finding a good local minimum due to the optimization landscape can be rougher.
>
> > The contribution ... them.
>
> * We updated the section to include more explanations to the evaluation criteria. In Table 1 in particular, we added a detailed description for each of the metrics in the table in caption.
>
> * We calculate mean for all $x_{uij}$s with known preferences of $i \succ j$. A larger value (and positive) of the parameter estimate shows that a model is able to separate item i from j to a better extend.
>
> * Consistency in our paper does not directly relate to ranking. In fact, when there are cycles in the directed acyclic graph induced by item pairwise preferences, no ranking that can be inferred. It motivates us to propose methods (M1 and M2) to rank items based on pairwise comparisons. We added more explanation in the revised manuscript on Pg 10.
>
> * Non-interacted items do have been used to build a criterion in previous studies. The well used and benchmarked metrics HR (Hit Ratio) is calculated based on non-interacted items (https://arxiv.org/pdf/1708.05031.pdf).
>
> > The paper ... field.
>
> The scope of our work has been highlighted in the Introduction. In fact, the seven SOTA methods used for benchmarking have covered a significant proportion of CF models developed for implicit feedbacks.
>
> > Details
>
> * Eq (5) is a unification of eq's in Section 4
> * Fixed
> * Fixed
> * Fixed
>
> > (Critical)
>
> See responses above.
>
> > (critical) Remove ...
>
> The paragraph is valuable as an explanation of how transitivity can be violated in real world. We merged the paragraph into its previous one.
>
> > (strengthening ...
>
> See responses above.
>
> > (critical) Clarify ...
>
> We added clarification.
>
> > (strengthen ...
>
> See responses above.

---

> > ### Comment · Editors_In_Chief · 2023-06-01
> > **further remarks**
> >
> > I thank the authors for their detailed and courteous response. I must apologize for some of my initial remarks, which were inappropriate:
> >
> > * I overinterpreted "While respecting antisymmetricity [...]". In fact, this paragraph is limited to the effect of item $i$ on the evaluation of item $j$ and does not discuss the effect of other displayed items on the comparison of item $i$ with item $j$. I think the new version of the paper is less subject to such over-interpretation, but the paragraph could still be improved.
> > * I originally read the curves in Figure 4 poorly. Clearly, with a small amount of missing data, SAD achieves a significantly lower error in both Sim1 and Sim2. The authors should consider removing some curves from Figures 4c and 4d to make them easier to read.
> > * Given the relationship of the paper to implicit feedback and learning to rank, I was expecting older SOTA approaches like *Weighted Regularised Matrix Factorisation* and *Collaborative Less-Is-More Filtering*, but these approaches may be outdated, and CLIMF is limited to binary feedback, while the current paper focuses on pairwise comparisons, which allows manipulation of more diverse feedback.
> >
> > However, I still have two mid-level concerns and one main concern.
> >
> > Mid-level concerns
> >
> > * I still think that Eq. (10) gives a better intuition of the difference between the new approach and BPR. In fact, this equation highlights the weighting effect of $\tau_{hi}$. Note that reviewer aiYy expected a model with a similar form (an extended version of your model where $\tau_{hi}\tau_{hj}$ is replaced by $<\tau_{hi},\tau_{hj}>$ ; see weakness 1.b of reviewer aiYy). Note also that Eq. (10) does not change the role of $\tau_{hi}$ in the model, so any comment about $\tau_{hi}$ in Eq. (10) also applies to $\tau_{hi}$ in Eq. (8).
> > * In the experiment on real data, the mean metric remains unclear in the paper (I understand it only thanks to the authors' reply). Please consider providing a formula in the paper.
> >
> > Main concern
> >
> > I now realize that the experiments are done on pairwise comparisons derived from ratings. Therefore, these comparisons are transitive. So if the proposed model has a better accuracy in the experiments, it's not due to its ability to model non-transitive relations. This is worrisome since the main motivation of the paper and the model is this capacity.

---

> > > ### Author Response · Authors · 2023-06-03
> > > **To further remarks**
> > >
> > > > I thank the authors for their detailed and courteous response. I must apologize for some of my initial remarks, which were inappropriate: ...
> > >
> > > Thanks for the comments. We apologize for the initial confusions.
> > >
> > > > Mid-level concerns
> > > >
> > > > * I still think that Eq. (10) gives a better intuition of the difference between the new approach and BPR. In fact, this equation highlights the weighting effect of $\tau_{hi}$ . Note that reviewer aiYy expected a model with a similar form (an extended version of your model where is $\tau_{hi} \tau_{hj}$ replaced by $\langle \tau_{hi}, \tau_{hj} \rangle$ ; see weakness 1.b of reviewer aiYy). Note also that Eq. (10) does not change the role of $\tau_{hi}$ in the model, so any comment about $\tau_{hi}$ in Eq. (10) also applies to $\tau_{hi}$ in Eq. (8).
> > > > * In the experiment on real data, the mean metric remains unclear in the paper (I understand it only thanks to the authors' reply). Please consider providing a formula in the paper.
> > >
> > > * Thanks for bringing up the commonality between Eq (10) and the parameterization from reviewer aiYy. There is actually a subtle difference between Eq (10) and modeling the $\tau_{hij}$ in a low rank factorization form. When replying reviewer aiYy’s comment, we intentionally made the distinction to formulate $\tau_{hij} = \langle \alpha_{hi}, \beta_{hj} \rangle$. In Eq (10) however, the assumption is stronger, with $\alpha_{hi} = \beta_{hi} ​= \tau_{hi}$​. Independent of the subtlety, introducing an additional interaction matrix $\{\tau_{hij} \} \in \mathbb{R}^{m\times m}$ requires additional constraints to meet. For example, to make the original $\{ x_{hij} \}$ to be anti-symmetric, the interaction matrix has to be a symmetric matrix. We think the exploration of alternative parameterizations is out of the scope of current work. We will leave this topic in our subsequent investigations. We revised our manuscript to add the conversation on this topic to Discussion on Pg 12.
> > > * We added Eq (11) in the revised paper to clarify the mean metric. The percentage metric is also explained by newly added Eq (12). Thanks for the comment.
> > >
> > > > Main concern
> > > > I now realize that the experiments are done on pairwise comparisons derived from ratings. Therefore, these comparisons are transitive. So if the proposed model has a better accuracy in the experiments, it's not due to its ability to model non-transitive relations. This is worrisome since the main motivation of the paper and the model is this capacity.
> > >
> > > We thank the reviewer for the comment. We view the limited integer-valued ratings as a coarse manifestation of users’ internal preference. We agree that the ratings are transitive for unevenly rated items. However, exiting scoring system is limited in that many items can have collided ratings. In our experiments, we chose the unevenly rated items as ground truth to benchmark different models. This is the best we can do to provide evidence of whether our model is able to produce consistent predictions. For evenly rated items, there is no ground truth available to allow us to evaluate the performance.
> > >
> > > This topic touches one fundamental aspect in our existing rating system - using an integer valued score with limited range as a sole reflection of user’s preference. Traditional collaborative filtering models pander to this system, by assuming there is a linear ranking among items, and the ranking is dictated by a real number representing the preference strength $\langle \xi_u, \eta_i \rangle$. However, reality can hold evidence that violates the assumption. For example, Bin Li et al. show the ubiquity of a user providing very different rating scores on closely correlated items, producing self-contradictions ( https://link.springer.com/article/10.1007/s11280-012-0161-9). In addition to noisiness in user ratings, the observed self-contradictions manifest the collision between user’s mental model and the rating system.
> > >
> > > SAD model proposed in our paper relies on pairwise comparisons, which is a more general mental model compared with linear rating systems. Our innovation lies in that we provided a novel methodology to mathematically parameterize the mental model. We added the discussion on Pg 12 in revised manuscript. We hope our explanations help resolve your concern.
> > >
> > > Our new revisions are highlighted in green in updated manuscript.

---

### Review · Reviewer_WWns · 2023-05-09

**Summary Of Contributions:**

This paper proposes a novel tensor decomposition based model for collaborative filtering based on implicit feedback. The proposed tensor decomposition is named sliced anti-symmetric decomposition (SAD). In contrast to standard methods that rely on vector representations of users and items, SAD forms a three-way tensor by introducing an additional latent vector for each item, thereby capturing higher order interactions between user-item pairs. The newly introduced latent vector per item allows for the notion of users having a non-linear mental model when evaluating models, as opposed to a linear mental model assumed by existing approaches. Experiments are performed on several synthetic and real-world datasets and the proposed method is compared with seven existing methods.

**Audience:**

Yes

**Claims And Evidence:**

No

**Requested Changes:**

1. Pg. 1, second paragraph, presents the statement "It is therefore more natural to assume that non-interacted items are a combination of the ones that users do not like and the ones that users are not aware of." Are the authors claiming this or re-stating this? If the authors are re-stating this, the original source needs to be cited. If not, please see if your experimental results agree to this statement.

2. $d_{uij} \\in \\{-1, 1\\}$ is used to represent if a user prefers the $i$-th item over the $j$-th or not. Could that authors clarify if the context of +1 or -1 here? That is, is +1 used for preference and -1 for otherwise?

3. Pg. 2 has the sentence "The model used in BPR assumes that among the observed entries in D, the probability that the u-th user prefers the i-th item over the j-th item can be modeled as a Bernoulli distribution". Can the authors please cite the source?

4. In (2) do we assume $x_{ui} > x_{uj}$? Is $x_{uij} > 0$. Please clarify.

5. The second set of non-negative item vectors here are denoted as $\tau_i$ for item $i$, can the authors give some insight onto what values could the entries of $\\tau_i$ take?

6. Pg. 3 has teo sentences that slightly contradict each other -- "Compared to traditional collaborative filtering methods
using matrix factorizations, tensor decompositions have received less attention in this field." and "Recently, tensor decomposition methods have been used to build recommendation systems using information from multiple sources." Could the authors please re-phrase?

7. Pg. 4 -- notation: vectors and scalars are denoted by lowercase letters and uppercase letters are reserved for matrices and tensors. This makes it hard for the user to identify the context in the text. The authors could consider simplifying notation by using lower case for scalars, lowercase boldface for vectors, uppercase boldface for matrices, and overline uppercase for tensors. E.g., $u$, $\\mathbf u$ $\\mathbf U$, and $\\overline{\\mathbf{U}}$.

8. It would be beneficial to see the first order slice of three-way tensor X corresponding to user $u$ as a graphical representation. Maybe show a cube (three-way tensor) and highlight the first order slice.

9. In Pg. 5, the notation used in (7) for SAD is commonly used for CP-decomposition. Is it possible to change the notation? If not, please clarify that in this paper, the notation in (7) is reserved for SAD.

10. Figures 1 and 2 appears way ahead of its corresponding text. Please move it closer to the corresponding text. Also, can we look at quantitative number for Figures 1 and 2? Maybe MSE?

11. Use \eqref in the first line of section 4.3 for 2 and 3.change (Equation 4) to (4) in the second paragraph.

12. (8) consists of two $\\tau$s -- $\\tau_i$ and $\\tau_j$. Where do they come from? It is not clear how we got to (8). Please provide clarification.

13. The text claims that SAD learns the value of T from data but that is not the case in experimental studies. That is, T is set explicitly and not learnt from data. Please re-phrase to be consistent with experimental studies.

14. Figure in Pg. 7 is not names. The title is missing. Should it be Figure 3? Many notations in this figure needs to be explained. For example, what are $p$, $p_{\\theta}$, what is the relationship between $D$ and $X_{true}$? Please clarify.

15. The figure in Pg. 7 corresponding to Sim1 sets T=1 and for this case, SAD must simplify to BPR, however, the performances of SAD (with T=1) and BPR are not the same. Why so? On the other hand in Sim 2, T is not equal to 1, so the behavior of SAD and BPR are expected to be different, but they appear similar, especially near convergence. Why? Please clarify.  Also please label the vertical line in (b)
as "true sparsity".

16.Please provide a pseudo-code of the proposed algorithm. This pseudo-code would contain (9) in it.

17. In section 5, please provide the dimensions of T. Please clarify how the hyper-parameters (learning rate, weight of L2 regularization, etc.) were selected? The number of latent factors k is set to the "true value". How can k be estimated in practice?

18. In Table 1, please define match, per user M1 and M2 in the title. Provide references next to the model names, that are compared against.

19. The dimension of the latent factor in real-world datasets is set to 500. Why? Can we cross-validate to find a dimension that works well?

20. This paper proposes pairwise comparisons for evaluation, namely, M1 and M2. Can the authors also report results on standard metrics, if any?

21. The discussion in section 7, mentions "When making recommendations,
the items forming a cycle can be selected as a group, ..." What does this mean? This is not mentioned anywhere in the maintext.


**Strengths And Weaknesses:**

Strengths:
1. The paper follows a good flow.
2. References appear to be sufficient.
3. Experiments conducted on multiple datasets and compared against seven existing approaches.

Weaknesses:
1. A lot of missing detail -- please see requested changes below.
2. The notation could be better -- see below.
3. Minor editing errors -- see below.

---

> ### Author Response · Authors · 2023-05-19
> **To reviewer WWns (to reduce characters, we labeled original numbered comments by C#)**
>
> > Strengths: *
> > Weaknesses: *
>
> Thanks for the summarization. See inline response to each requested change below.
>
> > C1
>
> It is a restatement. We added citation after the statement in the revised manuscript.
>
> > C2
>
> We added clarification and rewrote the statement.
>
> > C3
>
> We added citation after the sentence. Thanks for pointing out.
>
> > C4
>
> We do not make the assumption that $x_{ui} ​> x_{uj}$​(or $x_{uij}​>0$). Ignoring the subscript,
> we use $x$ to denote the natural parameter of a Bernoulli distribution with parameter $p \in (0, 1)$.
> Then $x=\log(p/(1−p))$. Its domain spans entire real number. We updated equation (2) to
> $x_{uij}​:= x_{ui} − x_{uj} \in \mathbb{R}$ in revised manuscript to highlight it.
>
> > C5
>
> The entries in $\tau_i$​ can take any positive values. In our work we assign an $l_1$ regularization
> to the entries to encourage them taking the value of 1. We revised the last sentence of Section 4.3
> by adding more explanation to what values $\tau_i$ can take.
>
> > C6
>
> Thanks for pointing out. We re-phrased the first sentence.
>
> > C7
>
> The notations used in the manuscript are mostly based on authors preferences as
> adapted to the literature on the topic. It is not rare to see notations that align with
> current manuscript, e.g. this [paper](https://jmlr.org/papers/volume15/anandkumar14b/anandkumar14b.pdf).
> To further help readers with the notation, we added additional hints in the revised
> manuscript to indicate the dimensionalities of mathematical entities. For example in
> Section 4 Pg. 4, $X$, $\Xi$, $H$ and $T$ are all explicitly labeled by their respective dimensions.
> We did the same in the rest of the revised manuscript.
>
> > C8
>
> Thanks for the suggestion. We added graphical representations for both $D$ and $X$ (Figure 1).
>
> > C9
>
> We changed to use curly brackets.
>
> > C10
>
> We replaced the two figures in the updated manuscript to make it closer to text.
> MSE were also added.
>
> > C11
>
> Fixed.
>
> > C12
>
> We rewrote the paragraph that introduced (8) to explain how $\tau$s were defined.
>
> > C13
>
> In Simulation we did set values of $T$ explicitly. This is because we want to examine the
> performance of our algorithm with true parameters known ahead. This is in contrast to
> the experimental studies that we did with real world data. We added more clarification
> at the beginning of simulation section.
>
> > C14
>
> We added the missing title of the figure. The explanation of notations
> were added to figure caption as well.
>
> > C15
>
> We explained the line in (b) in the figure caption. There are potentially several
> reasons that could cause the differences of performances of SAD and BPR in Sim1.
> One notable reason is that in SAD, even with $l_1$ regularization centered around 1,
> the entries of $\hat{T}$ can still take values other than 1. We chose to define sparsity
> as percentage of the entries in $\hat{T}$ with $|\hat{\tau_{hj}} - 1| < 0.05$. Those
> jittering around 1 can accumulate and contribute to the performance differences between the two models.
>
> In Sim2, there is actually significant difference between SAD and BPR near convergence.
> In Figure 4d in the revised manuscript, the solid lines (representing SAD) achieved a much
> lower error rate (Frobenius distance) compared with dashed lines (representing BPR).
>
> > C16
>
> We created pseudo-code in Algorithm 1. Thanks for the suggestion.
>
> > C17
>
> We added missing dimensions in Section 5. We found in
> the simulation study, changing the weights of $l_1$ and $l_2$ didn’t produce significant
> impact to the SGD algorithm. When applying to real world datasets, we did a grid
> search to find the values of those hyper-parameters (see Table 3 in Appendix C) that
> produced the best goodness-of-fit. For estimating $k$, cross-validation approaches
> can be used. We explicitly added this explanation in the second paragraph in
> Section 5 in the revised manuscript.
>
> > C18
>
> We added explanations of all metrics in the table to caption. The references to each model,
> together with package that implements it, are provided in Table 3 Appendix C.
>
> > C19
>
> In this paper we didn’t intend to find an optimal number of latent factors. Instead,
> we chose the same number of latent factors across comparing SOTA models such
> that they can be evaluated on a common ground. We delegate the estimation of the
> number of latent factors to future work. We added following sentences in revised
> manuscript to explain this.
>
> > C20
>
> The metrics reported in the paper actually were standard metrics being used in literature,
> namely HR (Hit Ratio). See for example in this [work](https://arxiv.org/pdf/1708.05031.pdf).
>
> > C21
>
> When transitivity properties no longer hold, cycles among items can be formed. Consider
> three items i, j and t. Transitivity implies if user u has $i \succ j$ and $j \succ t$ , then $i \succ t$ must hold.
> Violation of transitivity suggests $t \succ i$, therefore among the three items they form a
> preference cycle. We added the explanation in the discussion to resolve the confusion.

---

> > ### Comment · Reviewer_WWns · 2023-06-12
> > **Response to author edits**
> >
> > Thanks to the authors for incorporating the edits suggested. The paper looks good to me.

---

### Decision · Action_Editors · 2023-06-24

**Recommendation:** Accept with minor revision

**Comment:**

In this paper, the authors propose Sliced Anti-symmetric Decomposition (SAD) for collaborative filtering based on implicit feedback. SAD extends the Bayesian Personalized Ranking (BPR) model by introducing one additional latent vector to each item. In this way, users could use nonlinear mental models when evaluating items, allowing the existence of cycles in pairwise comparisons. Finally, the authors demonstrate the efficiency of SAD in both simulated and real-world datasets.

The proposed SAD method is novel and interesting. After discussions, the reviewers are generally positive with this paper. However, there still exists one concern regarding the dataset used in the experiments, which is shown below. When preparing the revision, the authors need to provide more discussions, and more experiments if possible.

---
The experimental part is done on data that do not contain the properties that the proposed model aims to capture. This section would be much more convincing if using data containing partial ranking on items and therefore may contain non-transitive relations, for example [1,2] or more recent.

1] Yandex personalized web search challenge. https://www.kaggle.com/c/yandex- personalized-web-search-challenge, 2013.

[2] KDD Cup 2012 Track 2. http://www.kddcup2012.org/


**Audience:**

Yes

**Claims And Evidence:**

Yes